

# Distinct changes in carbon, nitrogen, and phosphorus cycling in the litter layer across two contrasting forest-tundra ecotones

**Authors:**

Frank Hagedorn[1+], Josephine Imboden[1], Pavel A. Moiseev[3], Decai Gao[1,4], Emmanuel Frossard[2], Daniel Christen[1], Kontantin Gavazov[1], Jasmin Fetzer[1,2+]

[1] *Forest Soils and Biogeochemistry, Swiss Federal Institute for Forest, Snow and Landscape Research WSL, Birmensdorf, Switzerland*
[2] *Dept. of Environmental Systems Science, ETH Zurich, Zürich, Switzerland*
[3] *Institute of Plant and Animal Ecology, Ekaterinenburg, Russia*
[4] *Qianyanzhou Ecological Research Station, Key Laboratory of Ecosystem Network Observation and Modeling, Institute of Geographic Sciences and Natural Resources Research, Chinese Academy of Sciences, Beijing, China*

[+] contributed equally

* Correspondence to: Frank Hagedorn (frank.hagedorn@wsl.ch)



## Abstract

At treeline, plant life forms and species are abruptly changing from low stature plants in the tundra to trees in forests. Our study assesses how this shift in vegetation affects the quality and elemental composition of the litter layer and consequently, the microbial processing and nutrient release during decomposition. We sampled litter layers along elevation gradients across conifer and broadleaf dominated treelines in the Russian subarctic Khibiny mountains and hemiboreal South Urals. In incubation experiments using microlysimeters at 5 and 15°C, we measured carbon (C) mineralization and the release of inorganic nitrogen (N) and phosphorus (P), reflecting net N and P mineralization. We also measured releases of dissolved organic C and N and analyzed the functioning and stoichiometry of microbial biomass. Results showed that the chemical characteristics of the litter layer fundamentally changed across both treeline ecotones. On average, C:N and C:P ratios decreased by 56 and 65% from tundra to forest, whereas lignin contents showed a 110%-increase. The consistent decrease in C:N:P ratios in the litter layer from tundra to forest was paralleled by pronounced increases in net N and P mineralization from the tundra towards the low elevation forest in both treeline ecotones. In contrast to net nutrient mineralization, C mineralization and the release of dissolved organic C and N did not change across both treeline ecotones. In microbial biomass colonizing the litter layer, C:N and C:P ratios decreased on average by 26 and 74% from tundra towards forest. The metabolic quotient ($qCO_2$) correlated positively with C:N and C:P ratios of the litter layer. In support, the substrate-use efficiency estimated by the microbial use of $^{13}$C labelled glucose-6-phosphate increased from the tundra to the forest with decreasing C:N:P ratios. In contrast, potential activity of a range of C-N-P-acquiring extracellular enzymes showed no consistent pattern. Overall, our results give evidence that the vegetation shift from tundra to forest is associated with an abrupt increase in net N and P mineralization in the litter layer, accelerating nutrient cycling and increasing N and P availability. Experimental warming by 10°C was less important for net N and P mineralization than litter composition. This implies that indirect effects of climatic warming through changes in plant community composition with treeline advances seem to be more important for soil N and P cycling than direct temperature effects.



## 1. Introduction

Treelines represent a striking vegetation boundary with a transition from low-stature tundra plants such as graminoids and dwarf shrubs to forest trees within short distance. Globally, the position of treelines is primarily confined to growing season temperature (Körner and Paulsen, 2004; Hagedorn et al., 2020). Climate warming led to advances of treelines to higher altitudes in many mountainous regions over the last century, but forest expansions often lag behind climatic warming (Hagedorn et al., 2019; Büntgen et al., 2022). One of the contributing mechanisms for the retarded treeline shift could be nutrient limitation (Sullivan et al., 2015; Gustafson et al., 2021), possibly induced by a small nutrient mineralization from nutrient-poor organic matter in tundra (Fetzer et al., 2024) at low temperatures (Parker et al., 2018; Wang et al., 2021).

Nutrients bound to organic matter are released during decomposition along the continuum from litter to strongly transformed soil organic matter (SOM). In the initial phase, plant detritus is mineralized to $CO_2$ and inorganic nutrient forms or converted into microbial biomass, releasing nutrients upon their death (Berg and McClaugherty, 2020). Release of inorganic nutrients with decomposition is regarded as net nutrient mineralization (e.g. Nadelhoffer et al., 1991; Brödlin et al., 2019). However, a small fraction of organic matter is also leached in organic form, transporting carbon (C) and organically bound nutrients into deeper soil horizons (Hagedorn and Machwitz 2007; Fetzer et al., 2022). Along with climatic controls such as temperature and moisture, decomposition processes and nutrient mineralization are driven by the quality and nutrient contents of litter (Aerts, 1997; Gavazov, 2010; Wang et al., 2021) as well as by the soil biota colonizing decomposing litter (Zheng et al., 2018; Liu et al., 2019). In the litter layer, decomposing organisms have to degrade recalcitrant compounds, such as lignin, tannins and melanins that are frequently interlinked with nutrient-rich and/or labile components (Adamzyk et al., 2019; Sainte-Marie et al., 2021). In addition, microbial communities are facing pronounced stoichiometric imbalances between their biomass and the plant residues they decompose but microorganisms possess various adaptation strategies to overcome nutrient limitation (e.g. Mooshammer et al., 2014; Manzoni et al., 2021): (1) To some extent, microbes can enlarge the C:nutrient ratio in their biomass but their plasticity in adjusting the stoichiometry of microbial biomass is generally considered limited due to homeostasis (Mooshammer et al., 2014). However, microbial species are found to have a wide range of element ratios (Fanin et al., 2017; Zhang and Elser, 2017; Camenzind et al., 2021) and organic material is colonized by distinct microbial communities (Kaiser et al., 2015; Solly et al., 2017a; Zheng et al., 2018). (2) Another strategy is the production of extracellular enzymes, which is coupled to the uptake of C and nutrients and the adjustment of nutrient turnover in microbial biomass (Spohn and Widdig, 2017). (3) Furthermore, microorganisms can regulate their element use efficiency by releasing nutrients or carbon exceeding their demand (Mooshammer et al., 2014). When nutrients are scarce, microorganisms lack the nutrients needed to produce microbial biomass, which leads to so-called "overflow respiration", lowering their carbon-use efficiency (CUE) (Manzoni et al., 2021). Mineralized nutrients are then immobilized or recycled in microorganisms (Siegenthaler et al., 2024). The switch from C or energy limitation to nutrient limitation is observed to occur at distinct threshold element ratios (TER) which are well established for C:N ratios (of around 20 on a molar basis; Mooshammer et al. (2014)). However, empirical TER are less certain for C:P, with reported molar ratios in forest litter varying greatly between 300 and 1700 (Moore et al., 2011) with values differing between types of organic material (Heuck and Spohn, 2016).

Since litter decomposition is largely controlled by plant species and local abiotic conditions (Gavazov, 2010; Joly et al., 2023), the abrupt change in plant species and environmental conditions across treelines impacts litter decomposition and consequently, nutrient release. *In situ* litter-bag studies with standardized plant leaf litter in the



sub-Arctic Scandes, the Tibetan Plateau, and the Urals give evidence of a slower litter decomposition in tundra
than in forests below treeline (Liu et al., 2016; Solly et al., 2017b; Parker et al., 2018; Wang et al., 2021), which
is related to both the lower quality of tundra litter (Zheng et al., 2018) and the less favorable microclimate in tundra
than under forest canopy (Kammer et al., 2009; Parker et al., 2018). Substantially less is known on the release of
nutrients from decomposing litter of the treeline ecotone. Generally, tundra plants growing in nutrient-poor soils
produce nutrient-poorer litter than in forests (Wang et al., 2021). In conjunction with the observed slower litter
mass loss, this strongly suggests that nutrient release from decomposing litter will be smaller in tundra than in
forests. This might contribute to the decline in nutrient availability from forests towards tundra (Mayor et al., 2017;
Fetzer et al., 2024), which in turn could reduce tree growth and forest expansion (Kammer et al., 2009; Hagedorn
et al., 2019).

Our study aimed to (1) provide a comprehensive assessment of how C, N and P release in organic or inorganic
forms during litter decomposition varies across forest-tundra ecotones and (2) to examine how microorganism
cope with changes in litter quality from forest towards tundra. We sampled litter layers along elevation gradients
ranging from the boreal forests to the tree-free tundra in the South Urals and Khibiny mountains (Kola Peninsula)
Here, treelines remained largely untouched from anthropogenic land-use and have advanced by 4 to 8 m in
elevation per decade due to climatic changes (Hagedorn et al., 2014; Moiseev et al., 2022). As the dominant treeline
species is coniferous (*Picea obovata*) in the metamorphic South Urals and deciduous (*Betula pubescens*) in the
plutonic Khibiny mountains, we assumed that sampled litter layers encompassed a strong gradient of organic
compounds and stoichiometry. In the laboratory, we conducted a 12 week-long microcosm study to measure
potential C mineralization and the release of inorganic and organic N and P at 5°C and 15°C. In addition, we
studied the responses of microbial functioning to the range of litter layer characteristics by (i) analyzing C:N:P
ratios in microbial biomass, (ii) measuring the activity of extracellular enzymes hydrolyzing organic C, N, and P
compounds, and (iii) determining the metabolic quotient (qCO$_2$) as well as the substrate-use efficiency by tracing
$^{13}$C-labelled glucose-6-phosphate (G6P). The added G6P contained sufficient P for detecting it in P leached from
the litter layer, allowing to quantify net P mobilization or immobilization from this labile organic P source.

We hypothesized that (1) the potential release of C, N, and P from the litter layer will increase from tundra to forest
due to an improving litter quality with decreasing C:N:P ratios. We expected that the increase in net N and P
mineralization would be greater than that of C, as released N and P are more strongly immobilized by
microorganisms in tundra than in forest litter due to higher C:N:P ratios. (2) We expected that microbial
functioning will adapt to the decreasing litter stoichiometry from tundra to forest by releasing less extracellular
enzymes, decreasing the C:N:P ratio of microbial biomass and by adjusting microbial substrate use efficiency by
respiring less CO$_2$ per unit biomass in the nutrient-richer litter layer in the forest. (3) Increasing temperature will
enhance the release of inorganic and organic C, N, and P, but to a different extent.

## 2. Material and Methods

### 2.1 Study sites and sampling

Releases of C, N, and P as well as microbial functioning were studied in the litter layer of forest-tundra ecotones
in two Eurasian mountain ranges, the Khibiny mountains on the Kola peninsula (67°N, 34°E) and on the Iremel
massif in the South Urals (54°N, 58°E) noted thereafter as S-Urals. In both regions, litter layers were collected
during the peak growing season (July) along two replicate elevation transects crossing the treeline ecotone reaching
from the treeless tundra to the forest. They covered approx. 150 m in elevation, ranging from 325 to 470 m a.s.l.



in the Khibiny mountains and from 1260 to 1405 m a.s.l. in the S-Urals. The transects in both mountain ranges were located on gentle slopes of less than 10°. While the parent material in the Khibiny mountains is plutonic rock that is rich in apatite, it is chlorite-illite-quartz shales in the S-Urals. Soil types are podzolized sandy Rankers in Khibiny, and loamy Cambisols in the S-Urals with pH values of 4.1 and 3.5 at 0-10 cm depth in Khibiny and S-Urals, respectively. The dominant tree species is a conifer, Siberian spruce (*Picea obovata Ledeb.*) in the S-Urals,

while in Khibiny a broadleaf species *Betula pubescens ssp. Czerepanovii* dominates. The ground vegetation of the tundra in both regions is dominated by dwarf shrubs such as *Vaccinium* species, *Empetrum*, *Betula nana*, mosses, and lichens. In the S-Urals, there are also grasses and sedges such as *Carex vaginata Tausch* and *Festuca igoschiniae Tzvelev* (Solly et al., 2017b). The forest at lower elevation consists also of open areas without tree canopy, which is dominated by up to 2 m tall herbs (*Polygonum bistorta L., Polygonum alpinum All.*) in the S-

Urals (Solly et al., 2017b), while in Khibiny a mix of ferns, mosses, and herbaceous plants along with dwarf shrubs (*Vaccinium* spp.) and *Empetrum* prevail. In both regions, the litter layer was sampled at three elevation levels across the forest-tundra ecotone: (i) 'tundra' without trees where shrubs, herbs, lichens, and mosses dominate; (ii) 'treeline' with the uppermost trees (> 2 m, 4 m on average) growing in clusters (tree crown cover <10%), and (iii) the 'forest' at lowest elevation with approx. 10 m tall trees and a tree crown cover of 38% (Solly et al., 2017b).

Within these elevation levels, the litter layer was sampled in randomly selected plots (1 m × 1 m) under the tree 'canopy' that is typical for the respective elevation level in terms of age class and height, and an adjacent subplot (1 m × 1 m) in the 'open land', a canopy-free area with shrubs and herbaceous plants in between tree clusters (distance to trunk 2.5–6 m) (Solly et al., 2017b). Each of these ecosystem types was sampled in three randomly selected plots that were distributed along horizontal distances of 2 km in Khibiny and 5 km in the S-Urals. Overall,

our sample set comprised five ecosystem types, replicated across two elevation transects in each of the two mountain regions: (1) tundra (2) treeline-canopy, (3) treeline-open land with tundra-type vegetation, (4) forest-canopy, and (5) forest-open land between forest patches. Samples were stored at 4°C until incubation in the laboratory.

**2.2 Litter C-N-P release experiment**

Microcosms were used to simultaneously quantify potential C mineralization, net N and P mineralization as well as DOC and DON release. Material from the litter layer (10 g of fresh sample cut into 2 cm × 1 cm pieces) was placed in filtration systems (Millipore Stericup, 250 and 500 mL) that were leached repeatedly to measure element release (Brödlin et al., 2019). To prevent clogging of the 0.45 μm durapore membrane filters of the microlysimeters, a glass fiber filter (GF 6 100, Hahnemühle FineArt Gmbh, DE, 80 g, 0.35 mm, inorganic binder)

was put on top of the filters. In addition, we wrapped the samples into nylon net filters (approx. 2 μm) which were tied up with a polyamide cord (1.3 mm) to avoid floating of litter material and to allow the replacement of microlysimeters when filters were clogged.

Following an initial leaching, litter layer samples were incubated for two weeks in a climate chamber at 15°C and leached on a weekly basis to precondition the litter samples (Canali and Benedetti, 2006). Afterwards half the

samples were transferred to a climate chamber at 5°C, while the remaining remained at 15°C to cover the soil temperature range during the growing season. Another 2 weeks later, we started the experiment, measuring the C, N, and P release repeatedly after 1, 2, 3, 4, 6, 8, and 12 weeks. The experiment comprised 60 litter microcosms (2 regions × 5 ecosystem types × 3 plots × 2 temperatures). Additional four samples without litter material served as controls, with two of them incubated in each climate chamber. Carbon mineralization was determined for the entire

week in between leaching cycles by placing the microlysimeters into airtight containers (1000 mL) and trapping



respired $CO_2$ in 25 mL of 0.5 M NaOH. The amount of $CO_2$ trapped was determined by immediately measuring the reduction in electrical conductivity calibrated by titration with HCl following $BaCl_2$ addition. The molarity of NaOH was adjusted during the course of the experiment.

For the leaching, 100 mL of nutrient solution similar to those found in organic layers (400 µmol $L^{-1}$ $CaCl_2$, 50 µmol $L^{-1}$ $K_2SO_4$, and 50 µmol $L^{-1}$ $MgSO_4$; Brödlin et al., 2019) were added to the microlysimeters. A glass disk was placed on top of litter samples to avoid floating of litter materials and to ensure homogenous wetting. After one hour, the microlysimeters were leached using a vacuum pump (EcoTech, Bonn, Germany) with a suction of 55 hPa. The volume of the leachate was determined gravimetrically.

### 2.3 Chemical analyses of leachates

In the leachates, we measured DOC and total dissolved nitrogen (TDN) with a TOC/TN analyzer (TOC-L, Shimadzu, Corp. Tokyo, Japan). $NH_4^+$ concentrations in leachates were analyzed colorimetrically using a flow injection system FIAS-400 and a UV/VIS Spectrometer Lambda 2S (Perkin-Elmer, Waltham, MA, United States). Ion chromatography was used to determine $NO_3^-$ concentrations (ICS 3000, Dionex, Sunnyvale, CA, United States). Phosphate concentrations were measured with the malachite green method (Ohno and Zibilske, 1991). Total dissolved P was determined as $PO_4^{3-}$ with the malachite green method after the extract was oxidized with ammonium persulfate dissolved in 0.9 M $H_2SO_4$ and autoclaved (Tiessen and Moir, 1993). Dissolved organic N and DOP concentrations were calculated by subtracting dissolved inorganic N or P from TN or TP, respectively.

### 2.4 Chemical analyses of litter layer

Litter layer C and N contents were analyzed by an Elemental analyzer (Euro EA3000, Euro Vector, Pavia, Italy). For P, K, Ca, Mg, and Mn in ground litter, samples were first digested in 8 M $HNO_3$ with 0.6 M HF in a microwave digestion unit (MW ultraCLAV MLS, Milestone Inc., Shelton, CT, USA). Then, total element concentrations were measured using ICP-OES (Optima 7300 DV, Perkin Elmer, Waltham, MA, USA). Lignin contents were determined by extracting 1 g of ground sample three times with 25 mL of hot water and once with cold water (extraction time 15 min each). The residues of the hot water-extracted samples were first extracted four times with 25 mL of ethanol. An aliquot was then hydrolyzed with 72% sulfuric acid and autoclaved. In the extract, the amount of acid soluble lignin (ASL) was measured with a photometer at 205 nm, while the contents of Klason lignin were determined gravimetrically after incineration of remaining litter materials at 550°C for 4 h.

Additionally, a subsample of ground litter was analyzed by Fourier-transformed infrared spectroscopy (FT-IR) using a Vertex70 FT-IR Spectrometer with High Throughput Screening Extensions by Bruker Optics (Massachusetts, USA) to identify major functional groups in litter (Kammer et al., 2009). The following bands were evaluated (Duboc et al., 2012): 2920 $cm^{-1}$ representing aliphatic C-H stretches; 1650 $cm^{-1}$: C-O of carboxylates and aromatic C=C vibration; 1515 $cm^{-1}$ for C=C of aromatic groups, 1270 $cm^{-1}$ and 1230 $cm^{-1}$ for benzoic acids and C-O of (aryl) esters and of phenolic groups (Tatzber et al., 2010).

### 2.5 Microbial biomass C, N, P, and enzyme activities

After the 12-week long incubation, microbial biomass C, N, and P in litter layers were estimated by the chloroform fumigation-extraction method ((Brookes et al., 1982; Brookes et al., 1985; Vance et al., 1987). Briefly, litter was split into two aliquots, equivalent to 0.75 g dry mass. One aliquot was fumigated with chloroform at 25°C for 24 h. Fumigated and non-fumigated litter subsamples were extracted with 15 mL of 0.05 M $K_2SO_4$ after shaking them for one hour with 200 rpm (Brookes et al., 1985). Organic C and total N concentrations in the extracts were then



determined with a Shimadzu TOC/TN analyzer (TOC-V, Shimadzu Corp., Tokyo, Japan). Microbial biomass C and N were calculated from the difference in extracted C and N between fumigated and non-fumigated litter, divided by the factor $k_{EC}$ = 0.45 (Vance et al., 1987) and $k_{EN}$ = 0.54 (Brookes et al., 1985), respectively. For microbial biomass P, fumigated and non-fumigated litter subsamples were extracted with 0.5 M NaHCO$_3$ (pH=8.5)

after shaking them for one hour with 200 rpm (Brookes et al., 1982). Then, inorganic P in the extracts was determined using the malachite green method. To correct the P fixation during the NaHCO$_3$ extraction, a spike of KH2PO4 equivalent to 25 µg P g$^{-1}$ soil was used. Finally, microbial biomass P was calculated from the difference of the two extracts divided by $k_{EP}$ = 0.40 (Brookes et al., 1982).

The potential activities of five extracellular enzymes in the litter layer were determined using fluorogenic substrates

according to the method described by Saiya-Cork et al. (2002). The enzymes and substrates were: β-glucosidase (EC 3.2.1.21) and β-xylosidase (EC 3.2.1.37) assayed with 4-methylumbelliferyl (MUB)-β-D-glucosidase and MUB-β-D-xylopyranoside, respectively, involved in C cycling; N-acetyl-glucosaminidase (EC 3.2.1.30) and leucine-amino-peptidase (EC 3.4.11.1) assayed with MUB-N-acetyl-β-D-glucosaminide and L-Leucine-7-amido-4-methylcoumarin hydrochloride (L-Leucine-AMC), respectively, involved in N cycling; Monoester phosphatases

(EC 3.1.3.2) assayed with MUB-phosphate, involved in P cycling. Before assaying soil enzyme activity, the concentration of fluorogenic substrates was optimized to ensure saturated substrate concentrations (German et al., 2011). Briefly, 0.5 g (dw equivalent) of moist litter samples was mixed with 100 mL sodium acetate buffer (pH = 5.0) and stirred on a magnetic plate for 1 min. 200 µL litter suspensions and 50 µL specific substrates fluorescently labelled with AMC or MUB were added into assay 96-well microplates. The AMC- and MUB- linked substrates

were used to measure leucine-amino-peptidase and all other hydrolytic enzymes, respectively. 200 µL litter suspensions, 50 µL buffer and 200 µL buffer, 50 µL MUB or AMC were added into quench and standard 96-well microplates, respectively. All microplates were incubated in the dark at 25 °C for 4 h, and then 10 µL of 0.5 mol L$^{-1}$ NaOH was added to terminate the assay. Fluorescence was measured using a microplate reader (Synergy H1, BioTek, Winooski, VT) at 365 nm excitation and 450 nm emission. The potential activities of litter extracellular

enzymes were presented in units of nmol g$^{-1}$ dry soil h$^{-1}$ and calculated using the formula described by German et al. (2011).

### 2.6 Substrate use efficiency by tracking $^{13}$C-glucose-6-phosphate

Following the 12-week long incubation, we quantified the mineralization of $^{13}$C labelled G6P ($^{13}$C$_6$H$_{11}$O$_9$PNa$_2$ × (H$_2$O)$_x$, by Omicron Biochemicals, Inc.) to determine substrate use efficiency (SUE). In the experiment, litter layer

material from both temperature incubations (2.4 g dry weight (DW)-equivalent) were transferred in 150 mL glass jars (Schott), and then 3 mL of a $^{13}$C-labelled G6P-solute containing 6.76 mg $^{13}$C were added to each sample, which corresponded to 0.64% of C and 5±0.6% of P in the litter sample. Control samples received the same amount of solute as water. For each measurement of $^{13}$C mineralization, the container with the samples were first flushed with CO$_2$-free air. Then, the jars were closed with a septum and gas samples of the headspace were taken after a set

incubation period (see below) with a 15 mL syringe and injected into pre-evacuated glass vials (12 mL volume, Exetainer gas testing vials; Labco Limited, High Wycombe, UK). Samples were incubated for 4 hours and 1 hour at 5°C and 15°C, respectively. The δ$^{13}$C values of respired CO$_2$ were measured by connecting the pressurized vials to a GasBench II (Delta PlusXL; Thermo Finnigan). $^{13}$C mineralization was determined immediately, 4, 24, 48, 72, 168 hours after adding $^{13}$C-labelled G6P. Regular weight measurements ensured a constant water content

during the experiment. An aliquot of the litter sample was then extracted with 0.05 M K$_2$SO$_4$ either before or after



fumigation with chloroform. In these extracts, the $^{13}$C was oxidized to $CO_2$ using $K_2S_2O_8$ that was measured as $^{13}CO_2$ as described above.

**2.7 Data analysis and statistics**

Net C, N, and P release was estimated by multiplying C, N, and P concentrations with the volume of the leachate. When concentrations were below detection limit, half this value had been used for the calculation. For the longer time periods towards the end of the experiment, we interpolated linearly between two measurements. The temperature dependencies of C, N, and P release were evaluated by calculating $Q_{10}$ values [1]:

$$Q_{10} = (k_2 / k_1)^{\frac{10}{(T2-T1)}}$$

[1]

where $k_1$ as the measured process rate (e.g. C mineralization) at temperature T1, here 5°C and $k_2$ as the process rate at temperature T2, here 15°C. $Q_{10}$ values could not be calculated for N and P release from tundra litter where element concentrations in leachates remained below detection limits.

The metabolic quotient $qCO_2$ was calculated as the ratio of respired C and microbial C in the 12$^{th}$ week at the end of the incubation experiment. Microbial SUE was estimated as

SUE = $^{13}C_{microbial}$ / ($^{13}C_{microbial}$ + $^{13}CO_2$)              [2]

where $^{13}C_{microbial}$ is the $^{13}C_{excess}$ in microbial biomass measured by CFE (e.g. Gao et al., 2021) and $^{13}CO_2$ the amount of respired $CO_2$ from the added G6P.

The net recovery of P in leached dissolved inorganic P (DIP) as compared to the amount of P added by G6P was calculated from a simple mass balance as


            Net DIP recovery = $(DIP_{t1} - DIP_{t0})/G6P_{added}$           [3]

with $DIP_{t1}$ and $DIP_{t0}$ being the amount DIP leached after and before the addition of G6P.

Data was statistically analyzed by fitting linear mixed effect models by the restricted maximum likelihood (lme
function of the nlme package (Pinheiro et al., 2021); R Version 3.6.2, R CoreTeam (2020). The model included the fixed effects region, elevation distance relative to the treeline, vegetation type (canopy or open land), and temperature as well as their 2-way interactions, while microcosm was the random effect. The corAR1 function was included in the model to account for repeated measurements per microcosm. Elevation distance relative to the treeline was used as a numerical variable to indicate the direction and position relative to the treeline with the 260    tundra being 50 m above treeline and the forest being 100 m below treeline. Response variables were log-transformed before the analysis. Residuals were tested for normal distribution using a Shapiro-Wilk normality test.

## 3. Results

**3.1 Litter layer characteristics**

Litter layer showed distinct chemical composition across the treeline ecotones in both regions. Total N and P contents strongly increased with decreasing elevations from the tundra to the forest (Table 1; $p_{Elevation} < 0.001$). For instance, P contents were 3.0 and 1.7 times greater in the forest than in the tundra in the Khibiny mountains and S-Urals, respectively. They were also greater under tree canopies than under open vegetation areas in the tundra



and in between tree clusters (Table 1; $p_{vegetation} < 0.001$). The litter layer of the two regions showed similar N
contents ($p_{Region}$ = n.s.), while P contents were about 60% greater in the Khibiny mountains with a plutonic bedrock
than in the S-Urals with a quarzitic bedrock ($p_{Region} < 0.001$; Table 1)). These differences between mountain regions
were particularly pronounced in the litter layer in the open areas. Element ratios showed analogous patterns. Molar
C:N ratios of litter layer decreased from 63 in the tundra to 24 in the forest under tree canopy and 37 in the open
areas ($p_{Elevation} < 0.001$). C:P ratios showed even stronger declines from the tundra to the forest than C:N ratios.
Contents of Ca, Mg, and Mn in the litter layer increased from tundra towards forest ($p_{Elevation} < 0.001$), while K and
Fe showed no consistent pattern (SI, Table S1).

**Table 1**: Selected chemical properties of the litter for the five ecosystem types (tundra, open and closed forest
patches at the treeline, open and closed forest patches in the established forest) along the elevational gradients at
Khibiny and the South Urals (S-Urals). Values given for tannins, aliphatic components and aromaticity are relative
measures derived from FT-IR spectra (Duboc et al., 2012; Figure S1). Statistical significances (P values) were
tested with linear mixed effects model (lme). Numbers in bold stand for statistically significant results ($p < 0.05$).

| Region | Elevation | Vegetation | $C_{tot}$ | $N_{tot}$ | $P_{tot}$ | Klason lignin | Tannins 1620 | Aliphatic 2920 | Aromaticity index |
|---|---|---|---|---|---|---|---|---|---|
| | | | g kg⁻¹ | g kg⁻¹ | mg kg⁻¹ | g kg⁻¹ | cm⁻¹ | cm⁻¹ | |
| Khibiny | Tundra | Open | 442 | 11.1 | 1052 | 388 | 1.62 | 1.40 | 1.00 |
| | Treeline | Open | 448 | 11.4 | 1054 | 434 | 1.66 | 1.39 | 0.97 |
| | | Tree | 476 | 18.1 | 1302 | 533 | 1.70 | 1.33 | 0.88 |
| | Forest | Open | 454 | 14.7 | 1588 | 469 | 1.62 | 1.31 | 0.93 |
| | | Tree | 454 | 22.1 | 1975 | 553 | 1.73 | 1.28 | 0.84 |
| S-Urals | Tundra | Open | 411 | 8.0 | 384 | 131 | 1.45 | 1.43 | 1.13 |
| | Treeline | Open | 415 | 8.6 | 498 | 171 | 1.43 | 1.39 | 1.10 |
| | | Tree | 427 | 19.1 | 1261 | 416 | 1.62 | 1.33 | 0.90 |
| | Forest | Open | 439 | 14.5 | 835 | 374 | 1.51 | 1.26 | 0.88 |
| | | Tree | 451 | 21.3 | 1481 | 528 | 1.69 | 1.35 | 0.85 |
| Significance | | | | | | | | | |
| Region | | | **<0.0001** | 0.21 | **<0.0001** | **<0.0001** | **<0.0001** | 0.52 | **0.0449** |
| Elevation | | | **0.004** | **<0.0001** | **<0.0001** | **<0.0001** | **<0.001** | **<0.0001** | **<0.001** |
| Vegetation | | | **0.02** | **<0.0001** | **<0.0001** | **<0.0001** | **<0.0001** | 0.22 | **<0.0001** |
| Region × Elevation | | | **0.02** | 0.40 | 0.06 | **<0.0001** | **0.02** | 0.75 | **0.022** |
| Region × Vegetation | | | 0.55 | 0.36 | **0.006** | **0.008** | **<0.0001** | 0.14 | 0.58 |
| Elevation x Vegetation | | | **0.0175** | 0.80 | 0.46 | **0.045** | 0.50 | **0.016** | 0.14 |

Klason lignin contents increased from tundra to the forest (Table 1), ranging between 13% in the tundra to 55%
under tree canopy in the forest of S-Urals ($p_{Elevation}$ and $p_{Vegetation} < 0.001$). The differences were more pronounced
in the S-Urals with litter from conifer trees in the forest than in the Khibiny mountains with broadleaf trees ($p_{Region}$
× Vegetation $< 0.01$). Measured lignin contents correlated closely with the absorbance of FT-IR spectra at wavelength
of 1515 cm⁻¹ ($R^2$=0.83***) and the aromaticity index ($R^2$=0.81***) (SI; Figure S1). These spectra indicated the
greatest share of aromatic C=C in litter in the forest ($p_{Elevation} < 0.001$) and under canopies ($p_{Vegetation} < 0.001$; Table
1). Absorbance at 1620 cm⁻¹ as an indicator for the abundance of C-O of carboxylates and aromatic C=C related
to tannins (Duboc et al., 2012) suggests a high share of these compounds under trees ($p_{Vegetation} < 0.001$, SI Figure
S1, Table 1). In contrast, absorbance at 2920 cm⁻¹ as a relative measure for aliphatic C-H stretches (Duboc et al.,





2012) was greatest under tundra vegetation and decreased significantly from tundra to the forest level ($p_{\text{Elevation}}$ < 0.001, Table 1; SI Figure S1).

### 3.2 Carbon, nitrogen, and phosphorus release

During the 12-week long incubation experiment, total C loss by mineralization and leaching at 15°C averaged 8.6% of the initial C stock in the litter layer (Figures 1 and 2; SI Figure S2). Carbon mineralization was the primary pathway of C-loss, accounting for 92% of the C loss from the litter layer. Less than 1% of the C stock in the litter layer was leached as DOC (Figure 2). The contribution of DOC leaching to total C loss from the litter layer varied, ranged from 7% under tundra vegetation to 20% under coniferous trees in the S-Urals ($p_{\text{Vegetation}}$ < 0.01; Figure 2). Carbon mineralization was greater in the S-Urals than in the Khibiny mountains ($p_{\text{Region}}$ < 0.001). Also, the effect of vegetation type depended upon the region (Table 2). While C mineralization was smaller in the litter layer under canopies of spruce trees than under open areas in the S-Urals, there was no difference between these vegetation types in the Khibiny mountains dominated by birch ($p_{\text{Region}\times\text{Vegetation}}$ < 0.001). Nevertheless, the difference in litter C mineralization between the two vegetation types was modest with on average 16% higher rates in the tundra than in the forest ($p_{\text{Vegetation}}$ < 0.001; Figs. 1 and 2). Carbon mineralization correlated negatively and most closely with lignin contents ($R^2$=0.55***) and showed a weakly positive correlation with C:N ratios ($R^2$=0.15*) and C:P ratios ($R^2$=0.30***) of the litter layer (Figure 3). No correlation existed between C mineralization and contents of Ca, K, Mg, and Mn in the litter layer.

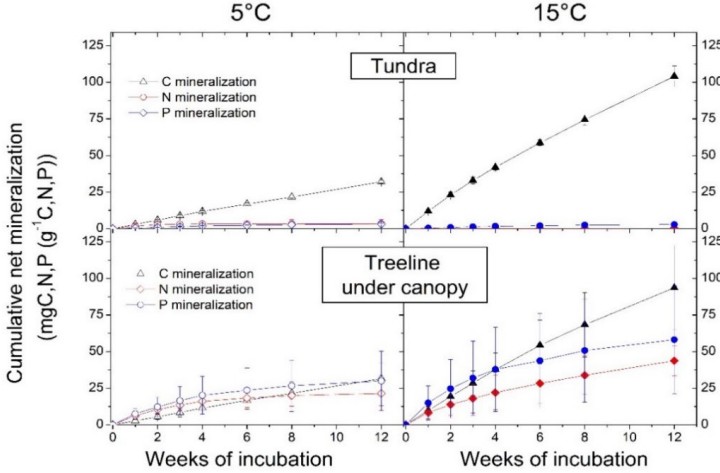


**Figure 1**: Cumulative potential net C, N, and P mineralization from the litter layer of the treeline ecotone in the South Urals at 5°C and 15°C during 12 weeks following a 3-week long preincubation at 15°C. Means and standard errors of three replicates. Cumulative values after 12 weeks of all vegetations types along the elevation gradient in the Khibiny mountains and South-Urals are shown in Figure 2.


The release of inorganic N and P strongly varied across the treeline ecotone (Figure 2). Here, we regarded the nutrients release from the purely organic litter layer to represent potential net nutrient mineralization (*sensu* Nadelhoffer et al., 1991; Weintraub & Schimel, 2003), distinguishing it from the apparent nutrient release measured from nutrient mass losses in litter bag studies (e.g. Moore et al., 2011). Potential net N mineralization from the litter layer was substantially smaller than C mineralization when related to their element contents (Figures





1 and 2). In both regions, net N mineralization rates remained close to the detection limit in the tundra and significantly increased at and beneath the treeline ($p_{\text{Elevation}} < 0.001$; Table 2). It also differed strongly between vegetation types ($p_{\text{Vegetation}} < 0.001$; Figure 2). In open areas, net N mineralization from the litter layer was negligible (<0.2% of its N content), while substantial amounts of N were mineralized from tree canopy litter,

reaching up to 6% of the N contents being mineralized in 12 weeks at 15°C at the lowest elevation. Overall, net N mineralization exhibited threshold-type relation with C:N ratios of litter materials (Figure 4); net N mineralization from litter ceased above C:N ratios of 35 (on a molar basis). Tundra litter had C:N ratios above this threshold element ratio, while tree canopy litter was below it.

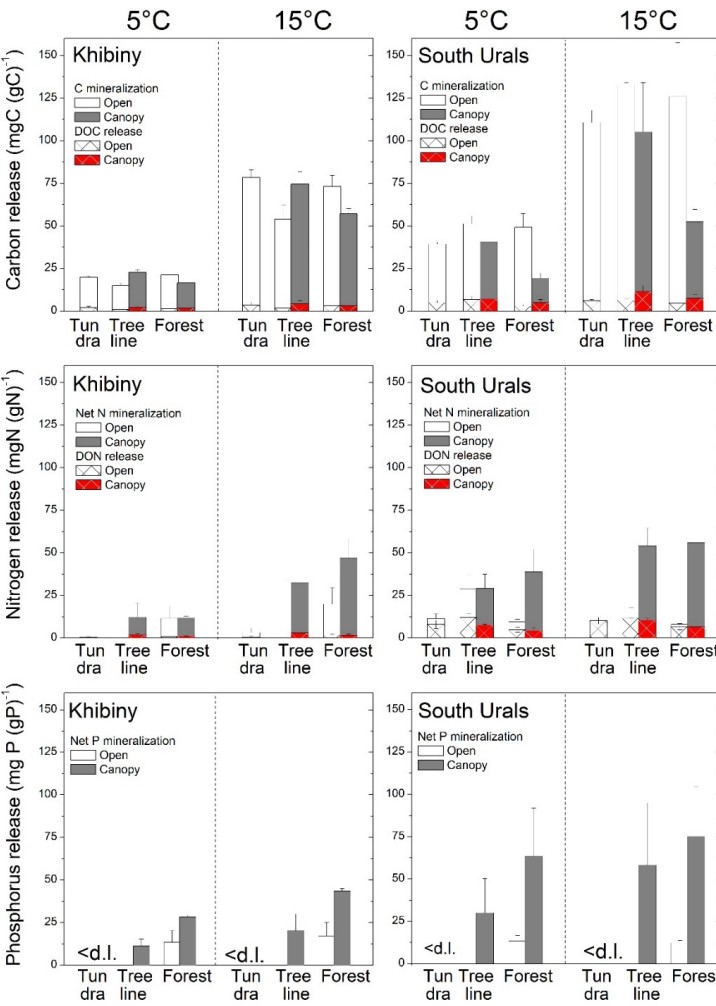

**Figure 2:** Cumulative net mineralization of carbon, nitrogen and phosphorus and release of organic carbon and nitrogen (DOC and DON) from the litter layer along elevation gradients across the treeline ecotone in the Khibiny mountains (left panels) and South-Urals (right panels) at 5°C and 15°C during 12 weeks of incubation. Means and standard errors of three replicates. Open land are areas without tree canopy consisting of a tundra vegetation in the tundra and at treeline, but forbs in the forest.



**Table 2:** Statistical significances (p values) of the linear mixed effects model (lme) testing effects of region, elevation, vegetation, incubation temperature, and their interaction on cumulative net C-N-P mineralization as well as DOC and DON release during 12 weeks. Numbers in bold stand for statistically significant results (p < 0.05).

|  | C mineralization | DOC release | DON release | Net N mineralization | Net P mineralization |
|---|---|---|---|---|---|
| Region | **<0.0001** | **<0.0001** | **<0.0001** | **<0.01** | **0.05** |
| Elevation | 0.013 | 0.36 | 0.51 | **<0.0001** | **<0.0001** |
| Vegetation | **<0.001** | **<0.01** | **0.035** | **<0.0001** | **<0.0001** |
| Temperature | **<0.0001** | **<0.001** | 0.09 | 0.89 | 0.26 |
| Region × Elevation | 0.07 | 0.19 | **0.001** | 0.04 | 0.45 |
| Region × Vegetation | **<0.001** | 0.41 | **0.01** | 0.82 | **0.01** |
| Elevation x Vegetation | **<0.001** | 0.56 | **0.043** | 0.26 | 0.26 |

The DON release correlated closely with DOC release among litter types (for cumulative releases: $R^2$=0.77***; Fig. 3). The average DOC:DON ratio of released DOM was 31. Amounts of DON released from the litter layer were greater under the tree canopy than in the open areas ($p_{\text{Vegetation}}$ = 0.035). However, due to the negligible net N mineralization from tundra litter and hence release of inorganic N, the contribution of DON to the total N release was greater in the tundra (60%) as compared to the tree canopy (13%) ($p_{\text{Vegetation}}$ < 0.001).

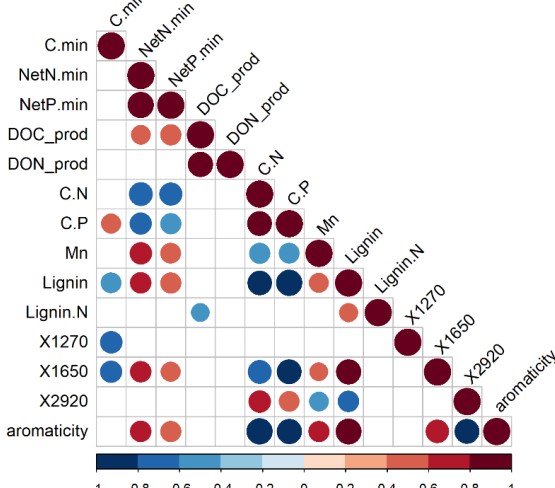

**Figure 3**: Pearson correlation coefficients among C mineralization, net N and P mineralization, release of dissolved organic C (DOC), dissolved organic nitrogen (DON) during 12 weeks and measures of the chemical quality of the litter layer across the forest-tundra ecotone in the two mountain ranges. min = mineralization, prod = production/release; C.N and C.P are litter C:nutrient ratios and Mn (manganese), lignin and nitrogen in lignin (Lignin.N) are litter composition parameters. X1270, X1650, and X2920 are values of the FT-IR spectrum at a given wavelengths that are proxies for phenolic groups, aromatic compounds, and aliphatic stretches, respectively.

Overall, potential net P mineralization correlated significantly with potential net N mineralization across litter types ($R^2$=0.70***; Figure 3). On average, net P mineralization was 20% higher than N mineralization when



normalized to element contents (Figures 1 and 2). In the litter layer under tundra vegetation, concentrations of P
in leachates from tundra litter remained consistently below the detection limit of 0.3 mg P $L^{-1}$ and thus, net P
mineralization was negligible. Concentrations of DOP were also mostly below detection limits in all litter layers.
Net P mineralization strongly increased towards the forest (Figure 2). The greatest net P mineralization occurred
in the litter layer under tree canopy in the forest reaching up to 8% of its P contents being mineralized during 12
weeks at 15°C ($p_{Elevation}$ and $p_{Vegetation} < 0.001$). The threshold for net P mineralization was at a molar C:P ratios of
approximately 1100 (Figure 4). Tundra litter had higher C:P ratios and thus net mineralization was negligible,
while tree canopy litter in the forest had lower C:P ratios and showed substantial net P mineralization.

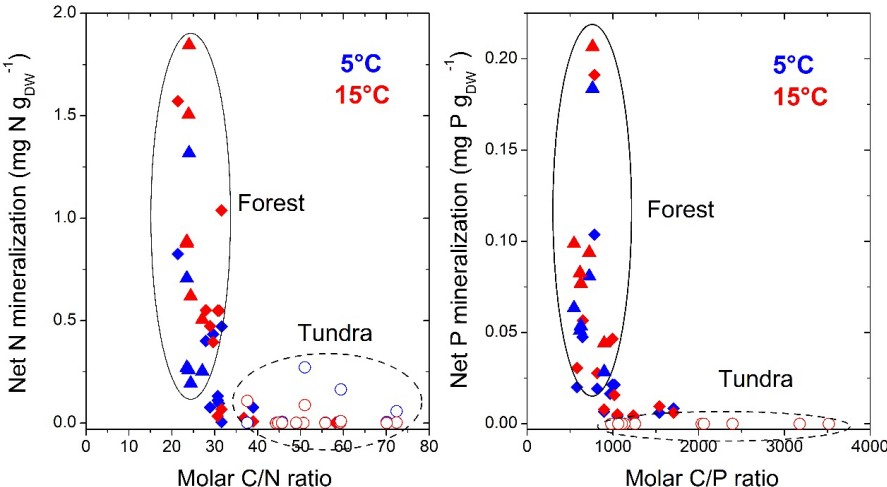

**Figure 4.** Relationship between the molar C/N and C/P ratios of litter materials to cumulative net N and P
mineralization during the 12 week-long incubation at 5°C and 15°C in the Khibiny mountains and S-Urals. Open
circles depict rates from tundra litter, closed triangles those from tree canopy litter in the forest, rhomboids those
at treeline in the two mountain regions.

### 3.3 Microbial biomass and extracellular enzyme activity

Microbial biomass in the litter layer was similar for the two regions but showed smaller pool sizes under tree
canopies than in open vegetation ($p_{Vegetation} < 0.001$, Table 3). Ratios of C:N and C:P in microbial biomass were
higher in Khibiny than in the S-Urals (Figure 5). In each region, they correlated significantly with the ratios in the
litter layer (Figure 5) and decreased from the tundra litter with high C:N:P ratios to the forest with lower ratios.
Microbial N:P ratios showed the same decline towards lower elevations as those of the litter layer ($p_{Elevation} < 0.001$;
Table 3). Molar N:P ratios in microbial biomass reached from 5 in the litter layer of the forest up to 20 in tundra.
The metabolic quotient ($qCO_2$) relating respiratory activity to microbial biomass in the 12th week of the experiment
decreased significantly with decreasing C:N and C:P ratios of the litter layer (Figure 5) and thus from tundra
towards forest ($p_{Elevation} < 0.001$). It was smallest in the litter layer under tree canopy ($p_{Vegetation} < 0.001$).
Extracellular enzyme activity was generally higher in the litter layer of the S-Urals than in Khibiny (SI, Figure
S3). Activities of enzyme hydrolyzing organic N (aminidase and peptidase) did not differ among elevation levels
and between vegetation types, for both mountain regions (Table 4). In comparison, phosphatase activity



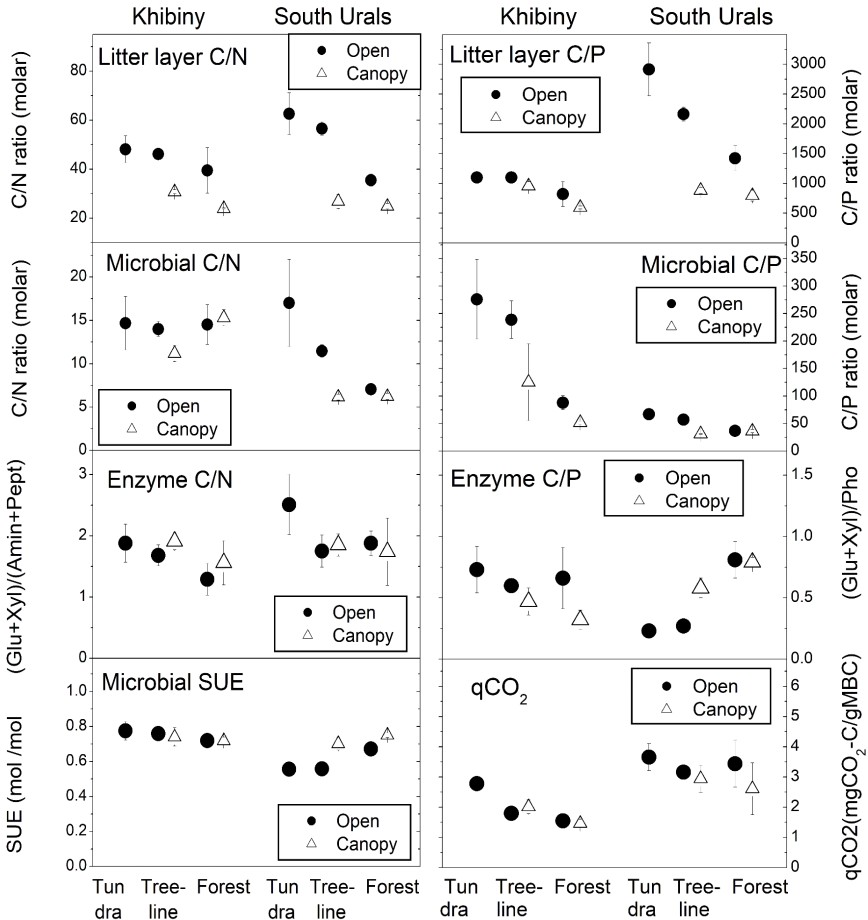


**Figure 5:** Molar C:N and C:P ratios in the litter layer and in microbial biomass, potential activity of element acquiring extracellular enzymes, metabolic quotient (qCO₂), and substrate use efficiency (SUE) across the forest-tundra ecotones in Khibiny and South Ural mountains. Tundra, treeline, and forest span distinct elevation gradients. Means and standard error of three plots after a 12-week long incubation at 15°C.

was higher in the P-poor litter layer of the open vegetation areas than in the P-richer litter layer under tree canopy in the S-Urals but showed no consistent pattern with elevation and among vegetation types in Khibiny ($p_{Region \times Vegetation} < 0.001$; SI Figure S3). Accordingly, the ratios of the activity of C:P acquiring enzymes of the litter layer (Figure 5) correlated negatively with the C:P ratio of the litter layer in the S-Urals ($R^2=0.44$**) but not in Khibiny. Ratios of the activity of C:N acquiring enzymes did not correlate with neither C:N ratios of the litter layer nor with those of microbial biomass (Figure 5).




**Table 3**: Statistical significances (p-values) of the linear mixed effects model (lme) testing effects of region, elevation, vegetation, temperature and their 2-way interactions on microbial biomass (MB)-carbon (C) and their C:N and C:P ratios as well as on the metabolic quotient ($qCO_2$) and substrate use efficiency (SUE). Numbers in bold stand for statistically significant results ($p < 0.05$).

| | MB-C | MB-C:N | MB-C:P | MB-N:P | $qCO_2$ | SUE |
|---|---|---|---|---|---|---|
| Region | 0.052 | **<0.0001** | **<0.0001** | **<0.0001** | **<0.0001** | **<0.0001** |
| Elevation | 0.25 | **0.0034** | **<0.0001** | **<0.0001** | **<0.0001** | **<0.0001** |
| Vegetation | **0.0004** | **0.0017** | **<0.0001** | **0.0004** | **<0.0001** | **<0.0001** |
| Temperature | **0.013** | **0.0036** | **0.0036** | 0.229 | **<0.0001** | 0.09 |
| Region × Elevation | 0.39 | **0.0028** | **0.0035** | **<0.0001** | 0.08 | **<0.0001** |
| Region × Vegetation | **0.0023** | 0.95 | 0.95 | 0.17 | 0.17 | **0.002** |
| Elevation x Vegetation | **0.0051** | **0.005** | 0.057 | 0.98 | 0.09 | 0.42 |


**Table 4**: Statistical significances (p-values) of the linear mixed effects model (lme) testing effects of region, elevation, vegetation, temperature and their 2-way interactions on extracellular enzyme activity. Data are shown in Figure S5. Numbers in bold stand for statistically significant results ($p < 0.05$).

| | Glucosidase | Xylosidase | Aminidase | Peptidase | Phosphatase |
|---|---|---|---|---|---|
| Region | **<0.0001** | 0.29 | **0.0005** | 0.87 | **<0.0001** |
| Elevation | **0.024** | 0.48 | 0.87 | 0.36 | **0.01** |
| Vegetation | 0.06 | 0.55 | 0.3 | 0.65 | 0.76 |
| Region × Elevation | 0.23 | 0.97 | 0.55 | 0.20 | **0.0001** |
| Region × Vegetation | 0.17 | 0.74 | 0.7 | 0.29 | **0.004** |
| Elevation x Vegetation | 0.34 | 0.40 | 0.98 | 0.06 | 0.28 |


### 3.4 Temperature effects

Carbon mineralization was significantly higher at 15°C than at 5°C (Figure 2; $p_{Temperature} < 0.001$) with a $Q_{10}$ value of 3.3±0.02 (Figure 6). Also, the release of DOC and DON was significantly increased by warmer temperatures ($p_{Temperature} < 0.001$), though their average $Q_{10}$ value was only 1.7±0.02. Temperature effects on net N

mineralization varied depending on litter type. In tundra type litter (open areas in the tundra and at treeline) with low net N mineralization rates, temperature had no effects. In forest litter, higher temperatures increased net N mineralization ($p_{Temperature} = 0.024$), resulting in a $Q_{10}$ value of 2.0±0.60 (Figure 6). Net P mineralization was not significantly affected by temperature. In forest litter with comparatively high mineralization rates, $Q_{10}$ values were 1.34±0.15. Microbial biomass and their N and P contents were rather independent from temperature.





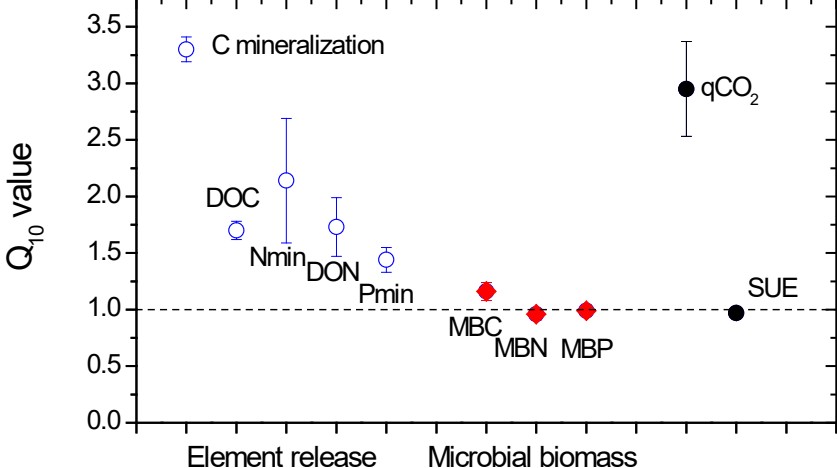


**Figure 6**: Temperature dependencies ($Q_{10}$ values) of the potential net mineralization of C, N, and P as well as the release of dissolved organic C and N (DOC and DON), C, N, and P in microbial biomass (MB), metabolic quotient ($qCO_2$), and Substrate-use efficiency (SUE) in litter layer incubated at 5 and 15°C. Averages and standard errors of all litter layer types in both mountain regions (n=30).

### 3.5 Tracking $^{13}$C labelled glucose-6-phosphate and substrate use efficiency

The addition of $^{13}$C labelled glucose-6-phosphate (G6P) led to an immediate $^{13}$C enrichment of respired $CO_2$ from the litter layer (SI Figure S4). During the following seven days, between 23% (litter under conifer trees from the

S-Urals) and 47% (tundra litter layer) of the added G6P were mineralized. Substrate use efficiency (SUE) increased from the tundra towards forest in the S-Urals, but not in Khibiny ($p$Region × Elevation < 0.002; Figure 5). Overall, SUE was significantly related to litter stoichiometry, correlating more closely with C:P than with the C:N ratios of the litter layer (Figure 7). Temperature did not affect SUE (Figure 6).

The G6P addition did not affect the net release of dissolved inorganic P (DIP) in the tundra litter layer with wide

C:P ratios although the amount of P added (0.175 mg P per sample) was more than twice as high as the highest observed P leaching rates from litter material (SI, Figure S5). In contrast, in the forest canopy litter with low C:P ratios, G6P addition increased the net DIP release from the litter layer (Figure 8). However, also here, the G6P-induced increase in net DIP release amounted to maximally 13% of added P. When relating the net DIP release to the amount of G6P mineralized estimated by the $^{13}$C tracing (22 to 47% of the added $^{13}$C), then the net DIP release

corresponded to maximally 50% of P potentially released from mineralized G6P. In tundra, net DIP release remained below 10% of mineralized G6P. Consequently, the greatest fraction of added G6P had been immobilized.





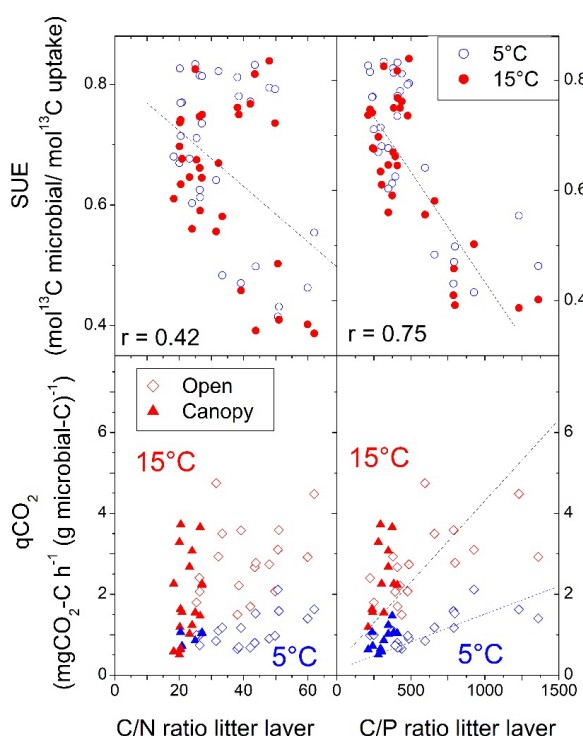

**Figure 7**: Relationship of molar C:N and C:P ratios in the litter and the metabolic quotient relating average C mineralization rates to microbial biomass as well substrate use efficiency (SUE) relating incorporation of $^{13}$C labelled glucose-6-phosphate into microbial biomass to the total amount of $^{13}$C label taken up by microorganisms.

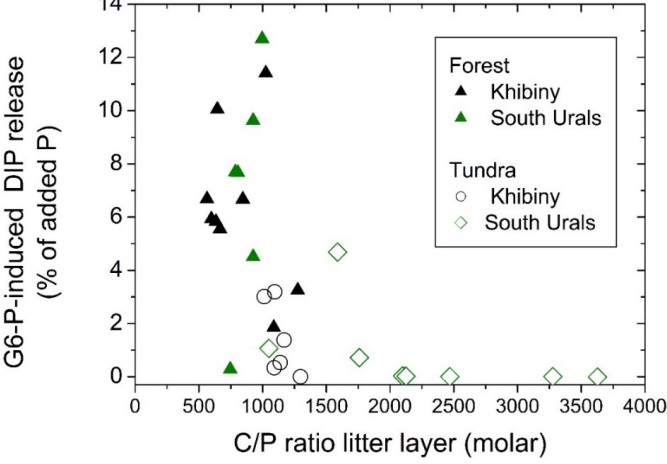

**Figure 8**: Relationship between the molar C:P ratio in the litter layer and the additional net release of dissolved inorganic P (DIP) following the addition of glucose-6-phosphate (G6P) in the tundra and forest of the Khibiny Mountains and the South Urals.





## 4. Discussion

### 4.1 Carbon, nitrogen, and phosphorus release across forest-tundra ecotones

Our study demonstrates that treeline ecotones represent a distinct boundary in litter quality and the C, N, and P release from decomposing litter layer with consequences for ecosystem N and P cycling across treelines. Consistent with our hypothesis, the composition of organic constituents and the stoichiometry of the litter layer strongly changed with the shift in plant life forms and species from tundra to forest (Table 1). These quality changes clearly impacted the processing of C, N, and P by microbial communities. Most strikingly, potential net N and P mineralization of the litter layer shifted from negligible rates in the treeless tundra to a pronounced nutrient mineralization in the litter layer under tree canopies at lower elevations. In contrast, C mineralization of the litter layer was surprisingly similar across the treeline ecotone, differing on average only by 16% between the litter layer in the tundra as compared to the forest (Figure 2). These small changes are consistent with the litter-bag study at treeline by Chen et al. (2018), observing that litter mass loss was less strongly affected by litter quality than by microclimate which was more favorable below than above treeline. In our study, we attribute the rather small changes in C mineralization to a cancelling out of litter quality characteristics of the litter layer across the forest-tundra ecotone. The litter layer in the tundra had smaller lignin contents, which is generally associated with rapid mineralization (Berg & McClaugherty, 2020). However, FT-IR analysis indicated that the tundra litter was also more aliphatic compared to forest litter, a characteristic typically linked to slower mineralization. In contrast to studies with tree litter (Hagedorn and Machwitz, 2007; Cornwell et al., 2008; Moore et al., 2011), C mineralization exhibited a positive correlation with C:N and C:P ratios. This could indicate that in the tundra with high litter C:N:P ratios microorganisms mineralized C in excess to acquire nutrients by the so-called "overflow respiration" (Mooshammer et al., 2014). However, it might also be related to cancelling out of the generally positive effects of decreasing C:N:P ratios by a lower quality of organic components. For instance, the litter layer under tree canopies had the lowest C:N:P ratio but at the same time also the highest contents of lignin that is more difficult to degrade. Despite the rather subtle changes in C mineralization across the forest-tundra ecotone, potential net N and P mineralization showed a threshold type increase from tundra to the forest canopy (Figure 4). The increased release of mineral N and P by more than a magnitude clearly exceeded a simple 'concentration or dilution effect' due to the litter layer having two to five times smaller N and P concentrations in the tundra than under the forest canopy. Consequently, N and P immobilization must have been contributing to the negligible net nutrient release from the litter layer in the tundra. This conclusion is supported by the complete net retention of P added with the easily mineralizable glucose-6 phosphate in tundra litter but to a lesser extent in tree canopy litter (Figure 8). Despite adding this easily available organic P source in an amount that exceeded the weekly net P mineralization from the P-rich tree litter by about one magnitude, mineralization of 50% of the added $^{13}$C-labelled G6P did not increase net DIP release. Consequently, and since it seems unlikely that mineralized and negatively charged $PO_4^{3-}$ becomes sorbed to negatively charged organic matter in the litter layer, it must have been immobilized, possibly in microbial biomass as observed in organic layer with low P availability (Siegenthaler et al., 2024).

In our incubation experiment, the Threshold Element Ratios (TER), at which net mineralization abruptly switches (Zechmeister-Boltenstern et al., 2015), were about 35 for C:N and 1000 for C:P for the litter layer of the two forest-tundra ecotones (Figure 4). These TER correspond to the critical ratios observed in mineralization studies of forest litter layer (e.g. Heuck and Spohn, 2016; Brödlin et al., 2019). For the C:N ratio, the observed TER is also in line with the one estimated from microbial nitrogen use efficiencies of microbial communities (Mooshammer et al., 2014). In our study, the TER for a net release from added G6P (Figure 8) coincided with the one of net P



mineralization again supporting that microbial immobilization causes the negligible P release (Figure 4). The TER ratios observed here, however, are slightly smaller than critical element ratios of net N and P loss estimated by litter bag studies (e.g. Moore et al., 2011; Manzoni et al., 2012). One obvious reason could be the bias by litter-bag studies through the export of nutrients through soil fauna and the leaching of dissolved organic nutrients

contributing to the nutrient mass loss. In our experiment with repeated leaching of litter layer material, the contribution of DON to total N release was particularly high for the low N litter layer in the tundra (60% on average). Consequently, N leaching losses occurred already above the TER for net N mineralization. We do not expect that DON released from the decomposing tundra litter improves the bioavailability of N in the tundra substantially as only a small fraction of DON consists of low-molecular compounds such as proteins or amino

acids that can be taken up by tundra plants (Weintraub and Schimel, 2005). Moreover, only a small fraction (15 to 40%) of dissolved organic matter leached from litter and forest floor layer is biodegradable (Hagedorn and Machwitz, 2007). Even when DON becomes mineralized, released inorganic N will become immobilized in the litter layer with very high C:N ratios as it was the case for phosphate released from mineralized G6P.

### 4.2 Flexible C:P ratios of microbial biomass

Microbial communities are generally regarded as homeostatic (Zechmeister-Boltenstern et al., 2015). In our study, however, C:N:P ratios of microbial biomass increased significantly from the forest towards the nutrient-poorer tundra along with increasing C:N:P ratios in the litter layer (Figure 5). This lends support for a stoichiometric plasticity of microbial biomass to overcome stoichiometric imbalance between resource material and decomposing communities (Fanin et al., 2017). The increase in microbial stoichiometry was more pronounced in the S-Urals

than in the Khibiny mountains and it was stronger for microbial C:P (molar ratio of 25 to 250) than for microbial C:N (6 to 15). One reason for the greater plasticity of microbial P could be the greater range in C:P ratios (600 to 2900) than in C:N ratios (24 to 63) in the litter layer (Figure 5), possibly resulting from a greater plasticity of plants for P (Yuan and Chen, 2009). Another reason could be a shift in the microbial community composition across the forest-tundra ecotone related to the changes in the quality and origin of the litter layer (Solly et al., 2017a). The

fungi/bacteria ratio in the litter layer decreased from the tundra towards the forest in the S-Urals (unpublished data, I. Djukic). Soil fungi have greater C:N:P ratios than bacteria (Mouginot et al., 2014) and fungal guilds show a twice as high span in their C:P ratios than in C:N ratios (Zhang and Elser, 2017). Consequently, microbial community shifts across the forest-tundra ecotone might strongly contribute to the observed plasticity of microbial C:N:P ratios. Also, isolates of saprobic fungi were found to possess a more flexible stoichiometry in response to

nutrient resources than previously thought (Camenzind et al., 2021). Fungi were also found to have a greater stoichiometric plasticity for C:P than for C:N ratios, possibly due to a modulating storage of polyphosphates. In our study, the microbial N:P ratio was rather homeostatic compared to the microbial C:P ratio, very likely due to the tight coupling of the synthesis of N-rich proteins with P-demanding ribosomal activity (Loladze and Elser, 2011).

### 4.3 Microbial functioning


Despite the plasticity of the C:N and C:P ratios in microbial biomass, there was still a stoichiometric imbalance between resource material and decomposing communities. Microbial functioning responded in various mechanisms to cope with this imbalance, but their relative importance differed between N and P and the two mountain regions. Consistent with the concept of TER which assumes that the C flow is controlled by the limiting

element (Zechmeister-Boltenstern et al., 2015), the amount of respired $CO_2$ as compared to microbial biomass ($qCO_2$) decreased with decreasing C-to-nutrient ratios and hence from the tundra to the forest. The greater C loss



from nutrient-poorer organic matter can be interpreted as 'overflow' respiration, where C in excess is disposed to acquire nutrients from more recalcitrant organic nutrient forms (Spohn, 2015). In agreement, we observed that the incorporation of $^{13}$C-labelled G6P in microbial biomass relative to the total amount taken up by microorganisms

decreased with increasing C:P and C:N ratios in the litter layer. This pattern reflects a flexible SUE of microbial communities with a smaller production of new microbial biomass per unit C when N and P are scarce (Manzoni et al., 2021). In the longer term, this may reduce the potential to sequester C.

Although the release of extracellular enzyme represents one of the mechanisms of microorganisms to overcome nutrient limitation (Manzoni et al., 2021), the only positive correlation of enzymatic activity with C:nutrient ratios

of the litter layer existed for phosphatase activity in the S-Urals (Figure 5). The lacking relation of phosphatase activity with litter C:P ratios in Khibiny might be attributed to generally higher P contents and thus C:P ratios below TER as well as to the pronounced changes of microbial C:P ratios across the forest-tundra ecotone in this region. Consequently, there was no need for microorganisms to invest in the production of phosphatases. In addition, the production of phosphatases might have been limited by the low nitrogen availability, especially in

tundra litter layer. Relations of enzyme activities from the N cycle with litter stoichiometry are less straight forward, as organic N molecules are also an important C source, and therefore their mineralization also depends on C-substrate preferences and/or C limitation of the microbial community (Zechmeister-Boltenstern et al., 2015).

### 4.4 Small temperature effects

While temperature expectedly stimulated C mineralization, effects of 10°C warmer temperatures on net dissolved

organic and inorganic N and P release were not significant. Possibly, for these net processes, resulting from the microbially driven production and immobilization – which are both temperature sensitive – have balanced out (Müller et al., 2009). The lacking temperature response of SUE agrees with the findings of Frey et al. (2013) suggesting that catabolic and anabolic activity of microorganisms are enhanced by temperature to a similar extent. The smaller effect sizes of the +10°C warmer temperatures as compared to litter type on net N and P mineralization

imply that indirect effect of climatic warming through changes in plant community composition – for instance by treeline advances – are more significant for soil N and P cycling in the litter layer than direct temperature effects.

### 4.5 Differences between treeline ecotones

Despite the regional difference with different bedrock types and tree species, there was a consistent increase in net N and P mineralization from tundra towards forest in both regions. Nonetheless changes in net C, N, and P

mineralization and microbial functioning across the forest-tundra ecotone were more pronounced in the S-Urals than in the Khibiny mountains. One likely reason is the different dominant tree species. While in the Khibiny mountains, broadleaf trees were dominating, coniferous trees formed the high elevation forest in the S-Urals. This caused a stronger contrast in the quality of the litter layer from tundra to the forest in the S-Urals than at Khibiny, where both deciduous shrubs and trees produce annual leaves above and below treeline. Another reason could be

the P-richer plutonic bedrock in the Khibiny mountains as compared to the quarzitic bedrock in the S-Urals. Consequently, the gradient in the C:P ratios of the litter layer from the forest to the tundra was less pronounced at Khibiny than in the S-Urals. However, more studies are needed to elucidate the modulating impact of inherent soil fertility and plants species composition on C and nutrient dynamics at treeline.

### 4.6 Ecosystem consequences

The strong increase in plant and litter stoichiometry and consequently in nutrient release from tundra to forest can possibly be attributed to the species-specific stoichiometric homeostasis of plant tissues (Elser et al., 2010). In the tundra, lichens and mosses in the tundra are characterized by lower nutrient concentrations compared to vascular



plants (Asplund and Wardle, 2013). In addition, a positive litter feedback is possibly contributing to the distinct increase in nutrient from tundra to forest (Fetzer et al., 2024). Tundra plants growing in soils with relatively low
N and P availability compared to the forest (SI Table S2), produce nutrient-poorer litter than in the forest. In turn, only negligible amounts of mineral nutrients are released from tundra litter during decomposition, which contributes to the lower N and P availability. Vice versa, N- and P-richer forest litter leads to higher net N and P mineralization and higher N and P availability in the forest. Potentially, this litter feedback has consequences for plant growth reducing plant vigor in the tundra but stimulating it in the forest - a conclusion supported by
fertilization experiments at treeline in the Scandes and in the Alps observing pronounced growth enhancements (Sveinbjörnsson, 2000; Sullivan, 2015) already at low fertilizer doses (Möhl et al., 2018). An analogous litter feedback promoting soil N cycling had been proposed for the enhanced shrub growth in arctic regions (e.g. Buckeridge et al., 2010).

Our laboratory experiment, however, estimated only potential net N and P mineralization, while *in situ* rates can
additionally be influenced by moisture patterns, freeze-thaw cycles (Gao et al., 2020; 2021), soil biota, or plant-mediated processes (Fetzer et al., 2024). Furthermore, N and P immobilized in microbial biomass and incorporated into soil organic matter becomes eventually released during mineralization of microbial necromass and/or SOM with lower C-to-nutrient ratios (see for N; Knops et al., 2002). However, regardless of whether N and P are released directly from the litter or following microbial recycling, the nutrient release per unit litter produced is much greater
in the forest than in tundra. This is supported by higher contents of available N and P in the soil beneath the litter layer in the forest compared to the tundra (SI Table S2). Also other treeline ecotones in temperate regions worldwide show increasing available N and P contents in the soil from tundra towards forest (Mayor et al., 2017).

## 5. Conclusions

Our study documents that treeline ecosystems represent a distinct boundary in the C, N, and P release from decomposing litter layer with consequences for ecosystem N and P cycling across treelines. Due to the strong changes in litter stoichiometry across treeline, net N and P mineralization from the litter layer was negligible in the tundra due to microbial immobilization, while large amounts of N and P were mineralized in the forest. Microbial functioning paralleled these changes, including an decreasing 'overflow' respiration of carbon in excess
with smaller litter C:N:P ratios in the forest compared to tundra. Additionally, C:nutrient ratios in microbial biomass decreased from tundra to forest. The observed plasticity was greater for microbial C:P than for microbial C:N ratios. The production of extracellular enzyme appeared less important. The temperature gradient of 10°C was less significant for net N and P mineralization than litter stoichiometry and therefore, indirect effects of climate warming by plant species shifts appear more important for N and P cycling than direct effects. We suggest that the
pronounced shift in net N and P mineralization across treelines leads to a positive 'litter feedback' which may contribute to the observed lag of treeline advances under climate warming as the 'slower' nutrient cycling and the resulting lower N and P availability may constrain the establishment of trees in tundra.

**Data availability**

All data are openly available at https://www.doi.org/10.16904/envidat.536



**Author contribution**

PM established sampling design in the field; FH, JF sampled litter layer; EF, and contributed to the conception of the study FH, JF, JI, DC, and DG designed and performed the laboratory study and analysis. JF, JI, and FH did the data analyses and the data visualization and wrote the first draft of the manuscript. All authors contributed to the interpretation of the findings and to the manuscript revision, and they read and approved the submitted version.

**Competing interest**

The corresponding author is Associate Editor in Biogeosciences.

**Acknowledgement**

We gratefully acknowledge the financial support for JF and FH by the Swiss National Science Foundation (SNF)
(project number 171171). We thank the WSL central laboratory (A. Schlumpf, K. v. Känel, J. Bollenbach, U. Graf, D. Pezzotta) and the WSL forest soil laboratory (A. Zürcher, B. Rahimi, N. Hajjar) for their support. We are also grateful to Philipp Baumann (ETH Zurich) for his support in measuring FT-IR spectra. This work was performed within the framework of the joint projects conceived by the Institute of Plant and Animal Ecology of the Ural Branch of the Russian Academy of Science (IPAE) and the Swiss Federal Institute for Forest, Snow, and Landscape
Research (WSL).

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
