# Peer review of "Distinct changes in carbon, nitrogen, and phosphorus cycling in the litter layer across two contrasting foresttundra ecotones"

_EGUsphere, 2024_

## Referee Comment (RC2)

General comment

The effects of global change on nutrient release from litter layers are certainly an actual and valid research objective in northern ecosystems The study presented by Hagedorn et al. contains interesting data, which seem to merit publication. However, the analysis of different ecotones with different vegetation will always lead to highly significant results. This means that the presentation of the data needs considerable improvement as the authors seem to be partly lost in data. They should consider shortening the text and removing some approaches, which do not add much information to the study.

Specific comments

The font is too small to allow easy reading of the PDF printout.
I would prefer continuous line numbering.
L38-41: Awkward statement! Microbial biomass and microbial residues also need to be mineralized for releasing nutrients.
L59: There is too much focus on overflow respiration in the current manuscript, which occurs mainly when high concentrations of low molecular weight organic substances are available to microorganisms. The authors should consider extracellular polymeric substances (EPS), fungal vacuoles and bacterial storage components, such as poly-hydroxybutyrate, as reasons for stoichiometric variability of soil microorganisms. Also, the presence or absence of Mn and Cu has often strong effects on lignin decomposition in litter layers.
L97-100: Awkward statement! Rephrase!
L143-144: I do not understand the reason for this initial leaching.
L152-153: Please, give the range of NaOH molarity.
L189: Brookes et al. (1985) and Vance et al. (1987) used 0.5 M K2SO4 for extracting mineral soil at a ratio of 1 to 4 (soil to extractant). The current authors extracted litter at a ratio of 1 to 20 (litter to extractant) with 0.05 M K2SO4. This deviation from the original references is based on previously published work in determining microbial biomass in litter, which should be cited in all fairness.
L191, L219, L226: remove "Corp", "Inc", and "Limited"!
L193: The kEC, kEN, and KEP values are not factors. The kEC value of 0.45 has been proposed by Wu et al. (1990), which should be cited.
L215: The formula should be given.
L243 and throughout the manuscript: The metabolic quotient is defined as basal respiration / microbial biomass C (Anderson and Domsch, 1990) and should not be used for the microbial use of a freshly added substrate.
Tables 1, 2, 3 and 4: The decimal numbers should be restricted to two, not in bold, non-significant numbers should be presented as NS.
L284-287: This is not a Results statement. Move to Materials and Methods or the Discussion section!
L306-309: It is impossible for me to get a clear information out of this poorly lay-outed Figure 1. The data of the endpoints should be given in a table.
L311-314: This is not a Results statement. Move to Materials and Methods or the Discussion section!
L325-329: Also, the layout of Figure 2 is poor. It does not make sense to adjust C, N, and P release to an identical scale. In addition, the figure contains excessive legends.
L326: I miss information on the DOC/DON ratio as quality index for the measurements.
L341-3??: I have doubts that these presentation of correlation coefficients is valid as the data are presumably not normally distributed as those presented in Figure 7.

L406-4??: Q10 values of MBC, MBN, and MBP should be removed.
L423-425: Figure 7 should be removed.
L426-428: It is not possible to distinguish the site-specific symbols using a greyscale print-out.
L435-436: Trivial statement! Remove!
L462-464: Awkward statement rephrase!
L490, L491, L502: "microbial biomass" not just "microbial"!
L579: Again, there is too much focus on overflow respiration. It is possible but cannot be clearly concluded from the current data.
L584-587: This statement is not a Conclusion. I miss a clear "take-home" message.

---

## Author Comment (AC1)

**Responses to Reviewer #1**

This is a nice paper about microbial C, N and P cycling in the litter layer of forest and tundra ecosystems. I have two suggestions and some rather minor comments.
**Response:** thanks a lot for your positive evaluation, the constructive comments, and the careful read. We incorporated all comments in the revised manuscript.

**Reviewer:** Concerning Fig. 8, I suggest adding a second panel that shows the percentage of respired $^{13}C$ (derived from glucose-6-phosphate). Based on this figure, the authors could discuss the loss of C in comparison to the loss of P.
**Response:** Thanks for the constructive suggestion. We have added a second panel to Figure 8 (now Figure 7 in the revised manuscript), clearly documenting that DIP release from added glucose-6-phosphate decreased with increasing molar C/P ratios despite an increase in 13C mineralization along this trajectory.

[Figure]

These findings were used to discuss P immobilization, for example: *"This conclusion is supported by the experiment tracking the fate of glucose-6-phosphate (G6P). While 20–50% of the $^{13}C$-labeled G6P was mineralized within 3 days in the litter layer, only a small fraction of the added P was released as phosphate (Figure 7)."*

**Reviewer:** It seems that the authors conclude that the main driver of the differences in microbial element cycling between forest and tundra is the difference in litter stoichiometry between these two ecosystems. This raises further questions. For example, about how the plants are able to acquire higher nutrient contents in the forest than in the tundra, and specifically how the forests are able to build up a larger N stock than the tundra ecosystems. I think that the (formation of the) larger N stock in the forest ecosystems is very relevant for the discussion of microbial N cycling in these ecosystems.
**Response:** Good comment and interesting aspect. Our additional data on N and P pools show that while the total N and P pools exhibit a modest increase from tundra to forest ecosystems,

the more pronounced changes occur in the inorganic forms of N and P (Table S2), which show a clear upward trend along this trajectory.

In the revised, the explanation for the stoichiometric patterns in the layer across treeline are addressed at the beginning as follows:

*"One reason for the pronounced change in litter stoichiometry is the species-specific stoichiometric homeostasis of plant tissues (Elser et al., 2010). For instance, lichens and mosses in the tundra typically have lower nutrient concentrations compared to vascular plants in forests (Asplund and Wardle, 2013). Plant-soil feedbacks may reicnforce the stoichiometric differences between tundra and forest vegetation, as the smaller C:N:P ratios in forest litter contribute to higher nutrient content in soil organic matter, thereby increasing nutrient availability (Fetzer et al., 2024). Additionally, tree roots and associated mycorrhizae enhance weathering and nutrient mining. While these processes primarily affects P rather than N, enhanced P availability— coupled with molybdenum mobilized in the rhizosphere—can promote $N_2$-fixation which is a critical mechanism for N accumulation in Arctic ecosystems (Rousk et al., 2017)."*

**Reviewer:** I would like to read some lines of text about this, including a short outlook about how this might change in response to global warming and what might happen with these element ratios when the treelines shift.

**Response:** We add a short outlook about the potential consequences of global warming by writing in the last sentence of the Conclusion: *"We suggest that the pronounced shift in net N and P mineralization across treelines leads to a positive 'litter feedback', where forest expansion driven by a warming climate will tighten the C-to-nutrient ratios in decomposing organic matter compared to tundra, which will in turn accelerate nutrient cycling and enhance nutrient availability. This potentially promotes the productivity of the advancing forest."*

As the discussion (and paper) is already relatively long, we tried to keep these additional aspects as short as possible.

**Further comments**

**L.50-60** the transport of nutrients (nitrogen and phosphorus) into decomposing litter by fungi is also be very important during the initial decomposition stages.

**Response**: Thanks, we added: *"[…]; (4) and fungi translocate nutrients from the soil into the nutrient-poor litter layer via their hyphae alleviating nutrient imbalances (Spohn and Berg, 2023)."*

**L. 59** Whether "overflow respiration" really occurs in soil microorganisms is debatable. It might be a process that can only be observed under extreme conditions in the lab.

**Response:** We agree with the reviewer and with reviewer#2 that "overflow respiration" is a misleading term and that we used it too often in the manuscript. Nevertheless, "overflow respiration" is used as a mechanistic term in key review publications on how microbes adjust their ecophysiology to low nutrient contents (Zechmeister-Boltenstern et al., 2015, Mooshammer et al., 2014; Manzoni et al., 2021). and we have to refer to it. We think that "overflow respiration" in natural systems can rather be understood as a simple balance between C and nutrients during microbial processing of litter, where C is lost as $CO_2$ while nutrients are recycled by microbial communities (when nutrient supply is limited). In the revised manuscript, it is removed from the Introduction and only presented once in the Discussion

**L. 71 and 81** It is not clear what exactly "litter quality" is. This term requires an explanation.

**Response**: We agree that litter quality (as soil quality) is a difficult term. We changed it too *"changes in stoichiometry and organic constituents in the litter layer"*

**L. 95** In hypothesis 1, the term "litter quality" requires an explanation.
**Response**: clarified to: *"due to decreasing C:N:P ratios."*

**L. 98** What is "microbial functioning"? This sounds a little awkward. I guess the authors mean microbial ecophysiology or something related.
**Response**: Thanks: We have changed the term to *"microbial ecophysiology"*

**L. 105** see above
**Response**: changed

**Section 2.1** It would be helpful to see a map and a few photos of the sites.
**Response**: We have added a map and few photos of the tundra and forest sites to the Supplementals (see end of the response letter). Adding it to the main manuscript would further increase the manuscript which is already long.

**L. 218/219** Was the glucose-6-phosphate uniformly labelled or was it only labelled in one C position?
**Response**: We added that the glucose-6-phosphate was uniformly labelled.

**L. 125** How was this done randomly. Please explain.
**Response**: We simply threw a ruler to sample the litter layer.

**Fig. 1** The label on the y-axis is not clear. This should be improved and further explained in the caption.
**Response**: We have revised the label on the y-axis:
*"Cumulative net mineralization of C,N, and P (mg C,N,P mineralized (glitter C,N,P)$^{-1}$)"*
Furthermore, in the Figure captions we now add an explanation: *"Amounts of mineralized C, N, and P are related to the masses of C, N, and P in the litter layer, respectively, to allow comparisons between the three elements."*

**Fig. 3 and corresponding text.** Please indicate whether these ratios are based on mass or number of moles. It is confusing that in some parts of the text, it is indicated that these are molar ratios while in other parts the authors simply refer to ratios.
**Response:** We used molar ratio in all our data evaluation and report is as molar when values are given. However, when we discuss the ratio in general, we did not add "molar" each time as it does not matter whether the ratio is given on a mass or molar basis.

**Fig. 8** I suggest to add a second panel to the figure that shows the percentage of respired $^{13}$C (derived from glucose-6-phosphate). This would allow the authors to discuss loss of C in comparison to loss of P.
**Response:** Thanks for the suggestion. We have added another panel, clearly documenting that while $^{13}$C mineralization from glucose-6-phosphate increased with the molar C/P ratio, DIP release decreased.

**Fig. 450/451** This sentence is not entirely clear. Specifically it is not clear what exactly "decreasing C:N:P ratios" refers to and what "effects" refers to.
**Response:** We have rephrased the sentence as follows: *"This could indicate that in the tundra with high litter C:N:P ratios microorganisms mineralized C in excess to acquire nutrients, a*

*mechanism that has been named as "overflow respiration" (Mooshammer et al., 2014). However, we rather relate the apparent positive relationship between C:N:P ratios and C mineralization to a changing composition in organic constituents along the same trajectory. For instance, while the litter layer under tree canopies had the lowest C:N:P ratio, it also contained the highest contents of lignin, which is more resistant to decomposition."*

**L. 455** replace "mineral" by "inorganic"
**Response**: changed

**L. 470** Microbial mineralization of what? I guess glucose-6-phosphate but it might be good to clarify this.
**Response**: clarified

**L. 474** It seems that rather the lab studies are biased (or artificial) because they exclude nutrient import and export.
**Response**: We agree that lab experiments can be more artificial. In our statement, we refer to the potential loss of organic N and P; litter bag studies do not allow to identify which form of the nutrient (organic or inorganic) had been lost. We rephrase the sentence:
*"One obvious reason could be that in litter-bag studies the export of nutrients through soil fauna and the leaching of nutrients in organic forms is not considered."*

**L. 484** is mineralized (not becomes mineralized)
**Response**: changed

**L. 502** Please replace "very likely" by "which might have been" (since this is rather speculative).
**Response**: changed accordingly

**L. 505** see above. What is "microbial functioning"?
**Response**: Changed to *"microbial ecophysiology"*

**L. 552** remove "in the tundra".
**Response**: Deleted and rephrased

**l. 556** This is not correct. While the litter has a lower nutrient content in the tundra than in the forest, 100% of the nutrients are released in both ecosystems after several decades
**Response:** We addressed this as follows:
We wrote: *"Furthermore, N and P immobilized in microbial biomass and incorporated into soil organic matter become eventually released during mineralization of microbial necromass and/or SOM with lower C-to-nutrient ratios (see for N; Knops et al., 2002)."*

**L. 572** remove "worldwide"
**Response**: removed

**S1 Study sites**

[Figure]

Figure 1: Study sites and photographs of the studied tundra and forest ecosystem in the Khibiny mountains and South Urals.

**References**

Asplund, J., and Wardle, D. A.: The impact of secondary compounds and functional characteristics on lichen palatability and decomposition, J. Ecol., 101, 689–700, https://doi.org/10.1111/1365-2745.12075, 2013.

Elser, J. J., Fagan, W. F., Kerkhoff, A. J., Swenson, N. G., and Enquist, B. J.: Biological stoichiometry of plant production: metabolism, scaling and ecological response to global change, New Phytol., 186, 593-608, https://doi.org/10.1111/j.1469-8137.2010.03214.x, 2010.

Fetzer, J., Moiseev, P., Frossard, E., Kaiser, K., Mayer, M., Gavazov, K., and Hagedorn, F.: Plant-soil interactions drive nitrogen and phosphorus dynamics in an advancing subarctic treeline, Glob. Chang. Biol., 30, e17200, https://doi.org/10.1111/gcb.17200, 2024.

Manzoni, S., Chakrawal, A., Spohn, M., and Lindahl, B. D.: Modeling microbial adaptations to nutrient limitation during litter decomposition, Front. For. Glob. Chang., 4, 1–23, https://doi.org/10.3389/ffgc.2021.686945, 2021.

Mooshammer, M., Wanek, W., Zechmeister-Boltenstern, S., and Richter, A.: Stoichiometric imbalances between terrestrial decomposer communities and their resources: Mechanisms and implications of microbial adaptations to their resources, Front. Microbiol., 5, 1–10, https://doi.org/10.3389/fmicb.2014.00022, 2014.

Rousk, K., Degboe, J., Michelsen, A., Bradley, R. and Bellenger, J.-P.: Molybdenum and phosphorus limitation of moss-associated nitrogen fixation in boreal ecosystems. New Phytol, 214, 97-107. https://doi.org/10.1111/nph.14331, 2017

Spohn, M., and Berg, B: Import and release of nutrients during the first five years of plant litter decomposition. Soil Biology and Biochemistry,176, 108878, https://doi.org/10.1016/j.soilbio.2022.108878, 2023.

Zechmeister-Boltenstern, S., Keiblinger, K. M., Mooshammer, M., Peñuelas, J., Richter, A., Sardans, J., and Wanek, W.: The application of ecological stoichiometry to plant-microbial-soil organic matter transformations, Ecol. Monogr., 85, 133–155, https://doi.org/10.1890/14-0777.1, 2015.

---

## Author Comment (AC2)

**Responses to Reviewer #2**

**General comment**
The effects of global change on nutrient release from litter layers are certainly an actual and valid research objective in northern ecosystems. The study presented by Hagedorn et al. contains interesting data, which seem to merit publication. However, the analysis of different ecotones with different vegetation will always lead to highly significant results. This means that the presentation of the data needs considerable improvement as the authors seem to be partly lost in data. They should consider shortening the text and removing some approaches, which do not add much information to the study.

**Response:** Thanks for the evaluation, the constructive comments, and the careful read. We incorporated all comments in the revised manuscript. We have removed one of the Figures (Fig. 7). Following the suggestions of the reviewer we have largely removed and rephrased the discussion of "overflow respiration. We agree with the reviewer that litter quality and its processing are likely to vary across ecotones. However, we believe our findings make two important contributions to the understanding of plant-soil interactions across treelines: (1) The results demonstrate that treelines act as a 'natural boundary' for microbial processing and nutrient cycling, and (2) they highlight that the higher nutrient release in forests compared to tundra has implications for vegetation dynamics—a factor often overlooked in treeline ecology.

**Specific comments**
The font is too small to allow easy reading of the PDF printout.
I would prefer continuous line numbering.
**Response:** We apologize but we formatted the manuscript according to the guidelines.

**L38-41:** Awkward statement! Microbial biomass and microbial residues also need to be mineralized for releasing nutrients.
**Response**: We followed the suggestion of the reviewer and rephrased the statement as follows:
*"In the initial phase, plant detritus is mineralized to $CO_2$ and inorganic nutrient forms or converted into microbial biomass (Berg and McClaugherty, 2020). Subsequently, nutrients can be released upon microbial residue decomposition."*

**L59**: There is too much focus on overflow respiration in the current manuscript, which occurs mainly when high concentrations of low molecular weight organic substances are available to microorganisms. The authors should consider extracellular polymeric substances (EPS), fungal vacuoles and bacterial storage components, such as poly-hydroxybutyrate, as reasons for stoichiometric variability of soil microorganisms. Also, the presence or absence of Mn and Cu has often strong effects on lignin decomposition in litter layers.
**Response:**
- **"overflow respiration"**: We agree with the reviewer that the term "overflow respiration" is misleading and unlikely to occur. Nonetheless, it is used in key review publications on how microbes adjust their ecophysiology to low nutrient contents in decomposed organic matter (Zechmeister-Boltenstern et al., 2015, Mooshammer et al., 2014; Manzoni et al., 2021). In the revised manuscript, we are using the term "overflow respiration" only once in the Discussion, where we think that this is actually not the underlying process.
*"This could indicate that in the tundra with high litter C:N:P ratios microorganisms mineralized C in excess to acquire nutrients, a mechanism that has been named as "overflow respiration" (Mooshammer et al., 2014). However, we rather relate the apparent positive relationship between C:N:P ratios and C mineralization to a changing composition in organic constituents along the same trajectory. For instance, while the litter layer under tree canopies had the lowest C:N:P ratio, it also contained the highest contents of lignin, which is more resistant to degradation."*

**- Low molecular weight organic substances:** We agree with the reviewer that such a study could profit from the analysis of LMW-substances but this would go beyond the scope of this study. We provide data on Mn in litter layers (SI Table 3)

**L97-100**: Awkward statement! Rephrase!
**Response:** Thanks, we rephrased the objectives as follows:
*"In addition, we studied the responses of microbial ecophysiology to the range of litter layer characteristics across the two treelines by (i) analyzing C:N:P ratios in microbial biomass, (ii) measuring the activity of extracellular enzymes hydrolyzing organic C, N, and P compounds, (iii) determining the metabolic quotient (qCO$_2$) as well as the use of $^{13}$C-labelled glucose-6-phosphate (G6P) by microorganisms, and (iv) quantifying net P mobilization or immobilization from the added G6P."*

**L143-144**: I do not understand the reason for this initial leaching.
**Response:** Clarified by writing: *"Following an initial leaching to standardize moisture conditions and remove nutrients released upon sample storage and processing, litter layer samples were incubated for two weeks in a climate chamber at 15°C and leached on a weekly basis to precondition the litter samples (Canali and Benedetti, 2006)."*

**L152-153**: Please, give the range of NaOH molarity.
**Response** : We provide the range as 0.05 to 0.1M NaOH

**L189**: Brookes et al. (1985) and Vance et al. (1987) used 0.5 M K2SO4 for extracting mineral soil at a ratio of 1 to 4 (soil to extractant). The current authors extracted litter at a ratio of 1 to 20 (litter to extractant) with 0.05 M K2SO4. This deviation from the original references is based on previously published work in determining microbial biomass in litter, which should be cited in all fairness.
**Response**: We have added the reference of Makarov et al. (2015) in the revised manuscript as suggested (Line 190)

**L191, L219, L226**: remove "Corp", "Inc", and "Limited"!
**Response**: These words have been deleted in the revised manuscript.

**L193**: The kEC, kEN, and KEP values are not factors. The kEC value of 0.45 has been proposed by Wu et al. (1990), which should be cited.
**Response**: We have changed "factor" to "extraction efficiency coefficient".
In addition, "Wu et al. (1990)" has been added in the revised manuscript.

**L215**: The formula should be given.
**Response**: Thank you for the suggestion. The formulas are added to the revised manuscript.

$$\text{"}Activity\ (nmol\ g^{-1}\ h^{-1}) = \frac{Net\ Fluorescence \times Buffer\ volume\ (ml)}{Emission\ coefficient \times Homogenate\ volume\ (ml) \times Time\ (h) \times Soil\ mass\ (g)}\text{[1]}$$

$$Net\ Fluorescence = \left(\frac{Assay - Homogenate\ control}{Quench\ coefficient}\right) - Substrate\ control \qquad \text{[2]}$$

$$Emission\ coefficient\ (fluorescence\ nmol^{-1}) = \frac{Standard\ fluorescence}{\left[\frac{Standard\ concentration\ (nmol) \times Assay\ volume\ (ml)}{Volume\ of\ standard\ (ml)}\right]} \qquad \text{[3]}$$

$$Quench\ coefficient = \frac{Quench\ control - Homogenate\ control}{Standard\ fluorescence} \qquad \text{[4]"}$$

**L243 and throughout the manuscript**: The metabolic quotient is defined as basal respiration / microbial biomass C (Anderson and Domsch, 1990) and should not be used for the microbial use of a freshly added substrate.

**Response**: We estimate the metabolic quotient for the litter layer, which had been in the field for extended time (at least 10 months). Therefore, the litter layer had been colonized by microbial communities before we started our incubation study. For the added glucose-6-phosphate, we use the term 'substrate-use efficiency'

**Tables 1, 2, 3 and 4**: The decimal numbers should be restricted to two, not in bold, non-significant numbers should be presented as NS.
**Response**: We reduced the decimal numbers, and effects of p>0.10 as NS. We still provide values of p<0.10 (not only restricted it to the commonly used p<0.05). Nevertheless, in the manuscript, we only speak from significant at p<0.05.

**L284-287**: This is not a Results statement. Move to Materials and Methods or the Discussion section!
**Response**: We moved it to the Introduction.

**L306-309**: It is impossible for me to get a clear information out of this poorly lay-outed
**Response**: We apologize for the low quality in the pdf. The figure documents the net mineralization of C, N, and P during 12 weeks. It provides information about the temporal patterns, the differences between the elements and between the main sites (tundra and tree canopy). We have re-formatted the Figure.

[Figure]

**Figure 1**. The data of the endpoints should be given in a table.
**Response:** In the Legend, we write: *"Cumulative values after 12 weeks of all vegetations types along the elevation gradient in the Khibiny mountains and South-Urals are shown in Figure 2."*

**L311-314**: This is not a Results statement. Move to Materials and Methods or the Discussion section!
**Response:** We moved the reference Introduction.

**L325-329**: Also, the layout of Figure 2 is poor. It does not make sense to adjust C, N, and P release to an identical scale. In addition, the figure contains excessive legends.

**Response:** We apologize for the low quality in the pdf. We have removed some of the legends, changed the color code and increased line sizes.

In our opinion, it makes sense to use identical scale by referring to the masses of each element, which allows a comparison of net mineralization rates among the elements (sensu Weintraub & Schimel, 2003; SBB). For instance, a smaller net N than C mineralization implies that released N during decomposition is immobilized.

See revised Figure 2:

[Figure]

**L326**: I miss information on the DOC/DON ratio as quality index for the measurements.
**Response**: In the revised manuscript, we provide DOC/DON ratios by writing:
*"The molar DOC:DON ratio of released DOM ranged between 23.5 under tree canopy and 168 in the tundra ($p_{Elevation}$ < 0.001)."*
In the manuscript, we kept discussion about DOM short in order to avoid extending further the manuscript.

**L341-3??:** I have doubts that these presentation of correlation coefficients is valid as the data are presumably not normally distributed as those presented in Figure 7.
**Response**: In Figure 3, we now use the Spearman Rank correlation coefficient, which is independent from the distribution of data. In the mixed effect model we log-transformed all data, accounting for the non-normal distribution of data.

**L406-4??**: Q10 values of MBC, MBN, and MBP should be removed.
**Response:** We think that it is an important information that MBC, MBN, and MBP were not temperature sensitive.

**L423-425**: Figure 7 should be removed.
**Response**: We have removed Figure 7. Mineralization of [13]C from the added G6P is now presented in the former Figure 8 together with net released DIP (following suggestions of reviewer 1).

**L426-428 (former Figure 8)**: It is not possible to distinguish the site-specific symbols using a greyscale print-
out.
**Response**: we have redrawn the Figure:

[Figure]

**L435-436**: Trivial statement! Remove!
**Response:** While the changes of organic constituents and stoichiometry is an expected outcome of the study, we think that the statement is needed as the litter quality changes are the reasons for the changes in nutrient release across the treeline ecotone. In the revised manuscript, we add explanation for the stoichiometric differentiation as follows:
*"Consistent with our hypothesis, the composition of organic constituents in the litter layer changed and C:N:P ratios strongly decreased with the shift in plant life forms and species from tundra to forest (Table 1). One reason for the pronounced change in litter stoichiometry is the species-specific stoichiometric homeostasis of plant tissues (Elser et al., 2010). For instance, lichens and mosses in the tundra typically have lower nutrient concentrations compared to vascular plants in forests (Asplund and Wardle, 2013). Plant-soil feedbacks may reicnforce the stoichiometric differences between tundra and forest vegetation, as the smaller C:N:P ratios in forest litter contribute to higher nutrient content in soil organic matter, thereby increasing nutrient availability (Fetzer et al., 2024). Additionally, tree roots and associated mycorrhizae enhance weathering and nutrient mining. While these processes primarily affects P rather than N, enhanced P availability— coupled with molybdenum mobilized in the rhizosphere—can promote $N_2$-fixation which is a critical mechanism for N accumulation in Arctic ecosystems (Rousk et al., 2017).*

**L462-464**: Awkward statement rephrase!
**Response:** we have revised the section about P immobilization by discussing the G6P experiment more in detail:
*"This conclusion is supported by the experiment tracking the fate of glucose-6-phosphate (G6P). While 20–50% of the $^{13}$C-labeled G6P was mineralized within 3 days in the litter layer, only a small fraction of the added P was released as phosphate (Figure 7). Again, there was complete net retention of the added P from the easily mineralizable G6P in tundra litter, albeit to a lesser extent in tree canopy litter. Our findings could have been influenced by sorption of mineralized phosphate (Brödlin et al., 2019); however, this seems improbable in the purely organic litter layer, where negatively charged organic matter does not sorb negatively charged $PO_4^{3-}$. Thus, phosphate mineralized either from the litter layer itself or from the added G6P must have been immobilized, potentially within microbial biomass, as observed in organic layers with low P availability (Siegenthaler et al., 2024)."*

**L490, L491, L502**: "microbial biomass" not just "microbial"!
**Response:** changed

**L579**: Again, there is too much focus on overflow respiration. It is possible but cannot be clearly concluded from the current data.
**Response:** we have removed the term "overflow respiration". Instead we wrote:
*"Microbial ecophysiology paralleled these changes, including a more efficient use of organic matter by microorganisms during decomposition with smaller litter C:N:P ratios in the forest compared to tundra."*

**L584-587**: This statement is not a Conclusion. I miss a clear "take-home" message.
**Response:** we rephrased the sentence as follows:
*"The study highlights that litter stoichiometry has a greater influence on net N and P mineralization than a 10°C temperature gradient, suggesting that the indirect effects of climate warming through plant species shifts are more critical for N and P cycling than the direct effects of temperature increases."*

**References**

Asplund, J., and Wardle, D. A.: The impact of secondary compounds and functional characteristics on lichen palatability and decomposition, J. Ecol., 101, 689–700, https://doi.org/10.1111/1365-2745.12075, 2013.

Berg, B., and McClaugherty, C.: Decomposition as a process–some main features in plant litter: decomposition, humus formation, carbon sequestration (Cham: Springer International Publishing), 13–43, https://doi.org/10.1007/978-3-030-59631-6_2, 2020.

Canali, S., and Benedetti, A.: Soil nitrogen mineralization. In: Bloem, J., Hopkins, D.W., Benedetti, A. (Eds.), Microbiological Methods for Assessing Soil Quality. CABI, Wallingford, UK, pp. 23–49, 2006.

Elser, J. J., Fagan, W. F., Kerkhoff, A. J., Swenson, N. G., and Enquist, B. J.: Biological stoichiometry of plant production: metabolism, scaling and ecological response to global change, New Phytol., 186, 593-608, https://doi.org/10.1111/j.1469-8137.2010.03214.x, 2010.

Fetzer, J., Moiseev, P., Frossard, E., Kaiser, K., Mayer, M., Gavazov, K., and Hagedorn, F.: Plant-soil interactions drive nitrogen and phosphorus dynamics in an advancing subarctic treeline, Glob. Chang. Biol., 30, e17200, https://doi.org/10.1111/gcb.17200, 2024.

Makarov, M.I., Malysheva, T.I., Menyailo, O.V., Soudzilovskaia, N.A., Van Logtestijn, R.S.P., and Cornelissen, J.H.C.: Effect of $K_2SO_4$ concentration on extractability and isotope signature ($\delta^{13}C$ and $\delta^{15}N$) of soil C and N fractions, Eur. J. Soil Sci., 66, 417–426, https://doi.org/10.1111/ejss.12243, 2015.

Manzoni, S., Chakrawal, A., Spohn, M., and Lindahl, B. D.: Modeling microbial adaptations to nutrient limitation during litter decomposition, Front. For. Glob. Chang., 4, 1–23, https://doi.org/10.3389/ffgc.2021.686945, 2021.

Mooshammer, M., Wanek, W., Zechmeister-Boltenstern, S., and Richter, A.: Stoichiometric imbalances between terrestrial decomposer communities and their resources: Mechanisms and implications of microbial adaptations to their resources, Front. Microbiol., 5, 1–10, https://doi.org/10.3389/fmicb.2014.00022, 2014.

Rousk, K., Degboe, J., Michelsen, A., Bradley, R. and Bellenger, J.-P.: Molybdenum and phosphorus limitation of moss-associated nitrogen fixation in boreal ecosystems. New Phytol, 214, 97-107. https://doi.org/10.1111/nph.14331, 2017

Siegenthaler, M.B., McLaren TI, Frossard E, and Tamburini, F.: Dual isotopic ($^{33}P$ and $^{18}O$) tracing and solution $^{31}P$ NMR spectroscopy to reveal organic phosphorus synthesis in organic soil horizons, Soil Biol. Biochem., 197, 109519, https://doi.org/10.1016/j.soilbio.2024.109519, 2024.

Weintraub, M.N., and Schimel, J.P.: Interactions between carbon and nitrogen mineralization and soil organic matter chemistry in arctic tundra soils, Ecosystems, 6, 129-143, https://doi.org/10.1007/s10021-002-0124-6, 2003.

Wu, J., Joergensen, R.G., Pommerening, B., Chaussod, R., and Brookes, P.C.: Measurement of soil microbial biomass C by fumigation-extraction-an automated procedure. Soil Biol. Biochem., 22, 1167-1169, https://doi.org/10.1016/0038-0717(90)90046-3, 1990.

Zechmeister-Boltenstern, S., Keiblinger, K. M., Mooshammer, M., Peñuelas, J., Richter, A., Sardans, J., and Wanek, W.: The application of ecological stoichiometry to plant-microbial-soil organic matter transformations, Ecol. Monogr., 85, 133–155, https://doi.org/10.1890/14-0777.1, 2015.